# Androgen drives melanoma invasiveness and metastatic spread by inducing tumorigenic fucosylation

Qian Liu[1,2,3], Emma Adhikari[1,2,3], Daniel K. Lester[1,2,3], Bin Fang [4],
Joseph O. Johnson[5], Yijun Tian[1], Andrea T. Mockabee-Macias[1,3], Victoria Izumi[4],
Kelly M. Guzman[5], Michael G. White [6], John M. Koomen [4,7],
Jennifer A. Wargo [6,8], Jane L. Messina[9], Jianfei Qi [10] & Eric K. Lau [1,3] ✉

Melanoma incidence and mortality rates are historically higher for men than women. Although emerging studies have highlighted tumorigenic roles for the male sex hormone androgen and its receptor (AR) in melanoma, cellular and molecular mechanisms underlying these sex-associated discrepancies are poorly defined. Here, we delineate a previously undisclosed mechanism by which androgen-activated AR transcriptionally upregulates fucosyltransferase 4 (*FUT4*) expression, which drives melanoma invasiveness by interfering with adherens junctions (AJs). Global phosphoproteomic and fucoproteomic profiling, coupled with in vitro and in vivo functional validation, further reveal that AR-induced FUT4 fucosylates L1 cell adhesion molecule (L1CAM), which is required for FUT4-increased metastatic capacity. Tumor microarray and gene expression analyses demonstrate that AR-FUT4-L1CAM-AJs signaling correlates with pathological staging in melanoma patients. By delineating key androgen-triggered signaling that enhances metastatic aggressiveness, our findings help explain sex-associated clinical outcome disparities and highlight AR/FUT4 and its effectors as potential prognostic biomarkers and therapeutic targets in melanoma.

Although melanoma has not generally been considered as a sex hormone-responsive cancer, growing clinical observations and emerging biological studies highlight the significant role that sex appears to play in the biology and prognosis of this deadly cutaneous malignancy. For example, men diagnosed with advanced melanoma generally exhibit poorer clinical outcomes than women[1,2]. Moreover, in 2023, there are ~47% more estimated new cases and twice the mortality of

melanoma in men than in women in the United States (American Cancer Facts & Figures, 2023). Indeed, sex has emerged as a poorly understood but independent prognostic indicator for melanoma[1]. Recent studies have begun to delineate biological roles of sex hormones underlying these clinical discrepancies. For example, G protein-coupled estrogen receptor (GPER) signaling was reported to suppress tumor growth and to increase anti-PD-1 immune checkpoint blockade

[1]Department of Tumor Microenvironment and Metastasis, H. Lee Moffitt Cancer Center & Research Institute, Tampa, FL, USA. [2]Cancer Biology Ph.D. Program, University of South Florida, Tampa, FL, USA. [3]Molecular Medicine Program, H. Lee Moffitt Cancer Center & Research Institute, Tampa, FL, USA. [4]Proteomics and Metabolomics Core, H. Lee Moffitt Cancer Center & Research Institute, Tampa, FL, USA. [5]Analytic Microscopy Core, H. Lee Moffitt Cancer Center & Research Institute, Tampa, FL, USA. [6]Department of Surgical Oncology, MD Anderson Cancer Center, Houston, TX, USA. [7]Department of Molecular Oncology, H. Lee Moffitt Cancer Center & Research Institute, Tampa, FL, USA. [8]Department of Genomic Medicine, MD Anderson Cancer Center, Houston, TX, USA. [9]Department of Pathology, H. Lee Moffitt Cancer Center & Research Institute, Tampa, FL, USA. [10]Department of Biochemistry and Molecular Biology, University of Maryland School of Medicine, Baltimore, MD, USA. ✉e-mail: Eric.Lau@Moffitt.org

efficacy in a female melanoma mouse model[3], whereas sustained AR signaling was reported to promote melanoma aggressiveness[4,5] and resistance to targeted therapies[6]. Intriguingly, androgen and estrogen appear to elicit opposite effects in melanoma[7], although underlying molecular mechanisms are poorly studied.

Fucosylation, the post-translational modification of glycoproteins with the dietary sugar L-fucose, is crucial for immunological and organ developmental processes[8–10]. Fucose moieties can be conjugated onto proteins through distinct structural linkages (*e.g.*, α-(1,2/3/4/6) and O-linkages) by 13 fucosyltransferases (FUTs), which can determine the stability, behavior, and activity of target proteins[9]. Given the structural diversity and functional specificity of the fucose-protein linkages uniquely catalyzed by the individual FUTs, the FUTs can elicit significant tumor-promoting or tumor-suppressing functions when their expression is deregulated[11]. Indeed, in cancer, aberrant levels of fucosylated proteins have been reported[10], but their specific mechanistic roles in cancer pathogenesis are unclear. Aberrant structural subtypes of fucosylation (*e.g.*, core fucosylation) have been reported in melanoma[12,13], and we previously discovered that global fucosylation is deregulated during melanoma progression, impacting motility and RNA processing of melanoma cells[14,15] and altering crucial tumor:immunological interactions that are required for anti-tumor immunity[16]. Intriguingly, our recent work uncovered that global tumoral fucosylation is significantly lower in male compared to female melanoma patients, suggesting sex-associated divergence in fucosylation-regulated melanoma biology[16]. In line with this possibility, glycosylation was previously identified as an androgen-regulated process that regulates cell growth in prostate cancer[17]. However, sex hormone-regulated fucosylation in melanoma is, at present, completely undefined.

Here, we report—for the first time—on how sex-hormone-regulated fucosylation mechanistically contributes to the disparately poor outcomes observed in male melanoma patients. The mechanism that we delineate appears to help explain how androgen/AR signaling shapes melanoma malignancy, enhancing invasive and metastatic capacity by inducing tumorigenic fucosylation. We identified fucosyltransferase 4 (*FUT4*) as a key transcriptional target of AR that plays a crucial role in mediating androgen-stimulated invasiveness by disrupting cell-cell adhesion complexes in melanoma. Integrated proteomics analysis identified L1CAM as a key downstream effector fucosylated by FUT4, which is required for AR-FUT4-promoted melanoma metastasis. Single-cell level assessment of a melanoma patient tumor microarray revealed a robust correlation of transcriptionally active AR, fucosylated L1CAM, and loss of cell-cell junction complexes with stage IIB-III tumors in male patients, consistent with the contribution of this signaling mechanism to augment invasive capacity required for early stages of the metastatic cascade.

Based on our identification of this androgen- and fucosylation-regulated molecular mechanism underlying clinical discrepancies associated with male melanoma patients, our study provides compelling preclinical support for the concept of treating melanomas using AR antagonists (*e.g.*, those approved for treating prostate cancer[18]) as well as for the implementation of androgen- and fucosylation-based biomarkers for potential therapeutic stratification of melanoma patients.

## Results

### Melanoma cells express androgen-inducible and transcriptionally active AR

Analysis of a large-cohort TCGA skin cutaneous melanoma dataset revealed variable expression of *AR* at the mRNA level in ~88% of all patients, and interestingly, the mRNA expression levels of *AR* appear to modestly increase from primary to metastatic specimens in male ($p = 0.02$) and female (not statistically significant, likely due to lower specimen numbers) patients (TCGA_SKCM; $n = 473$ melanoma cases; Fig. 1a, b). Assessment of 11 patient-derived melanoma cell lines

showed that, while significantly lower compared to a prostate cancer cell line (LNCaP), 4 of the 11 cell lines (WM793, WM1366, IPC298, and A375) express detectable AR protein, regardless of mutation background or sex origin (Fig. 1c and Supplementary Fig. 1a). Acute stimulation of AR+ melanoma cells with 100 nM dihydrotestosterone (DHT) triggered ~200% upregulation of total AR protein, with concomitant and marked nuclear translocation of AR, indicating that melanoma-expressed AR is responsive to androgen stimulation (WM793 cells in Fig. 1d, e; WM1366, IPC298, and A375 cells in Supplementary Fig. 1b, c). In contrast, DHT treatment of AR- melanoma cells failed to induce AR expression (LU1205 and WM266-4 cells in Supplementary Fig. 1d). Androgen response region (ARR2)-containing promoter luciferase assays[19] confirmed that the DHT-induced AR in melanoma cells is also transcriptionally active (Fig. 1f). Taken together, these data demonstrate that consistent with classical AR activation and function, in melanoma cells, androgen-activated AR accumulates in the nucleus where it exhibits transcriptional activity.

### Androgen is required for melanoma proliferation and migration in vitro and in vivo

We sought to delineate if and how androgen-induced and transcriptionally active AR impacts melanoma biology, and thus, tumorigenic capacity. Culture of AR+ WM793 melanoma cells in steroid hormone-depleted medium (10% charcoal-stripped serum (CSS)) inhibited cell viability—an effect that was entirely reverted by DHT supplementation (Fig. 1g, *left*). Moreover, DHT treatment alone significantly enhanced the proliferation and motility of AR+ melanoma cells in vitro (Fig. 1g, *center and right*; Supplementary Fig. 1e, f). In contrast, DHT stimulation failed to rescue the viability of AR- LU1205 melanoma cells under steroid hormone depletion (Supplementary Fig. 1g). Consistent with these observations, the growth of SM1 mouse melanoma tumors was blunted in castrated C57BL/6 syngeneic male mice (Fig. 1h and Supplementary Fig. 1h). These findings show crucial roles of androgen-induced AR signaling that likely contribute to the pathogenesis of melanoma.

### Androgen/AR is a regulator of cellular fucosylation in melanoma

In prostate cancer, androgen-activated nuclear AR functions as a classical transcription factor with relatively well-characterized transcriptional repertoire, including genes such as *PSA*, *KLK2*, *FKBP5*, and *TMPRSS2*, the alteration of which promotes prostate tumor aggressiveness[20,21]. In light of (i) previous report highlighting glycosylation as a crucial androgen-regulated process that controls the viability of prostate cancer cells[17] and (ii) our recent finding uncovering the reduction of a major subtype of fucosylation in male versus female melanomas[16], we sought to assess if AR might transcriptionally control any fucosylation machinery genes. Among the 19 predominant fucosylation genes, we identified 4 genes (*FUT4*, *FUT1*, *SLC35C2*, and *FUK*) that contain putative AR binding sites within their 5'-promoter or 3'-UTR regions[22] (Fig. 2a). Of the 4 genes, androgen stimulation resulted in significant downregulation of *FUK* and upregulation of *FUT4*—suggesting that AR regulates *FUK* and *FUT4* expression in melanoma cells (Fig. 2b, *left*).

To understand the tumor-promoting versus tumor-suppressing functional implications of altered fucosylation in melanoma, it is important to appreciate how fucosylation is fundamentally regulated at the following 2 mechanistic levels: global substrate availability and structure-function. FUK, the most upstream regulator of the fucose salvage pathway, initiates the conversion of free cellular L-fucose to GDP-fucose. As a key rate-limiting enzyme in this pathway, up- or downregulation of FUK can significantly increase or decrease, respectively, GDP-fucose and therefore global cellular fucosylation levels without altering structural subtypes of fucosylation[14] (Supplementary Fig. 2a, blue dashed box). In contrast, FUT4 is 1 of 13 downstream FUTs that mediate conjugation of GDP-fucose onto specific

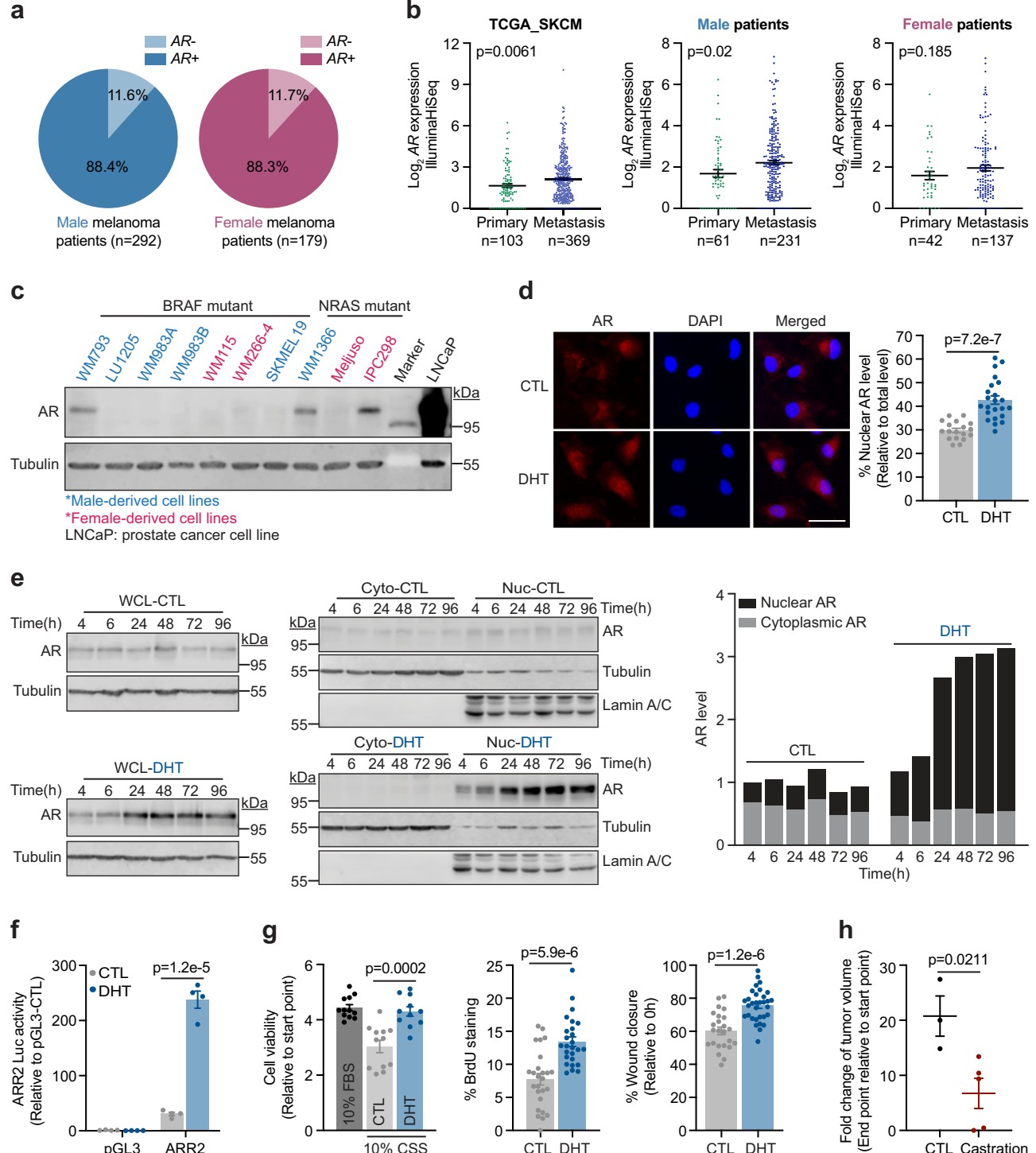

**Fig. 1 | Melanomas express androgen-inducible, transcriptionally active AR, which drives proliferation and motility.** *AR* mRNA levels in **a** male vs. female or **b** primary vs. metastatic melanoma tissues from the TCGA skin cutaneous melanoma dataset (TCGA_SKCM; *n* = 473). **c** Immunoblotting (IB) analysis of baseline AR protein levels across 10 human melanoma cell lines. LNCaP prostate cancer cell line serves as a positive control for AR expression. Uncropped blots in Source Data. **d** Immunofluorescence (IF) staining of AR protein in WM793 cells treated ± 100 nM dihydrotestosterone (DHT) for 8 h. CTL, *n* = 18 fields; DHT, *n* = 23 fields examined over 3 independent experiments. Scale bar = 50 µm. **e** Whole cell lysate (WCL; *left*) and subcellular fractionation (*center*) IB of AR protein in WM793 cells treated with vehicle (CTL; *upper*) or 100 nM DHT (*lower*) over 96 h. For subcellular fractionation blots, tubulin and lamin A/C indicate cytoplasmic (Cyto) and nuclear (Nuc) fractions, respectively. Stacked column chart (*right*) shows quantified subcellular localization of AR protein from the blots (*left and center*). Uncropped blots in

Source Data. **f** ARR2 luciferase reporter assay of WM793 cells treated ± 100 nM DHT for 48 h (*n* = 4 biologically independent samples). **g** (*left*) MTT assay of WM793 cells cultured in 10% fetal bovine serum (FBS) or 10% charcoal-stripped serum (CSS) ± 100 nM DHT for 4 days (10% FBS, *n* = 12; CTL, *n* = 12; DHT, *n* = 11 biologically independent samples). (*center*) BrdU staining (*n* = 26 fields examined over 3 independent experiments) and (*right*) scratch migration assay (CTL, *n* = 24 scratches; DHT, *n* = 32 scratches examined over 3 independent experiments) of WM793 cells treated ± 100 nM DHT for 48 h. **h** The fold-change of SM1 tumor volume in C57BL/6 mice at the end point (35 days after implantation). Mice were castrated at 1.5 weeks prior to injection (CTL, *n* = 3 mice; Castration, *n* = 5 mice). For **b, d, f–h**, data are presented as mean values ± standard error of the mean (SEM) and *p*-values are calculated by two-sided Student's *t*-test. Source data are provided as a Source Data file.

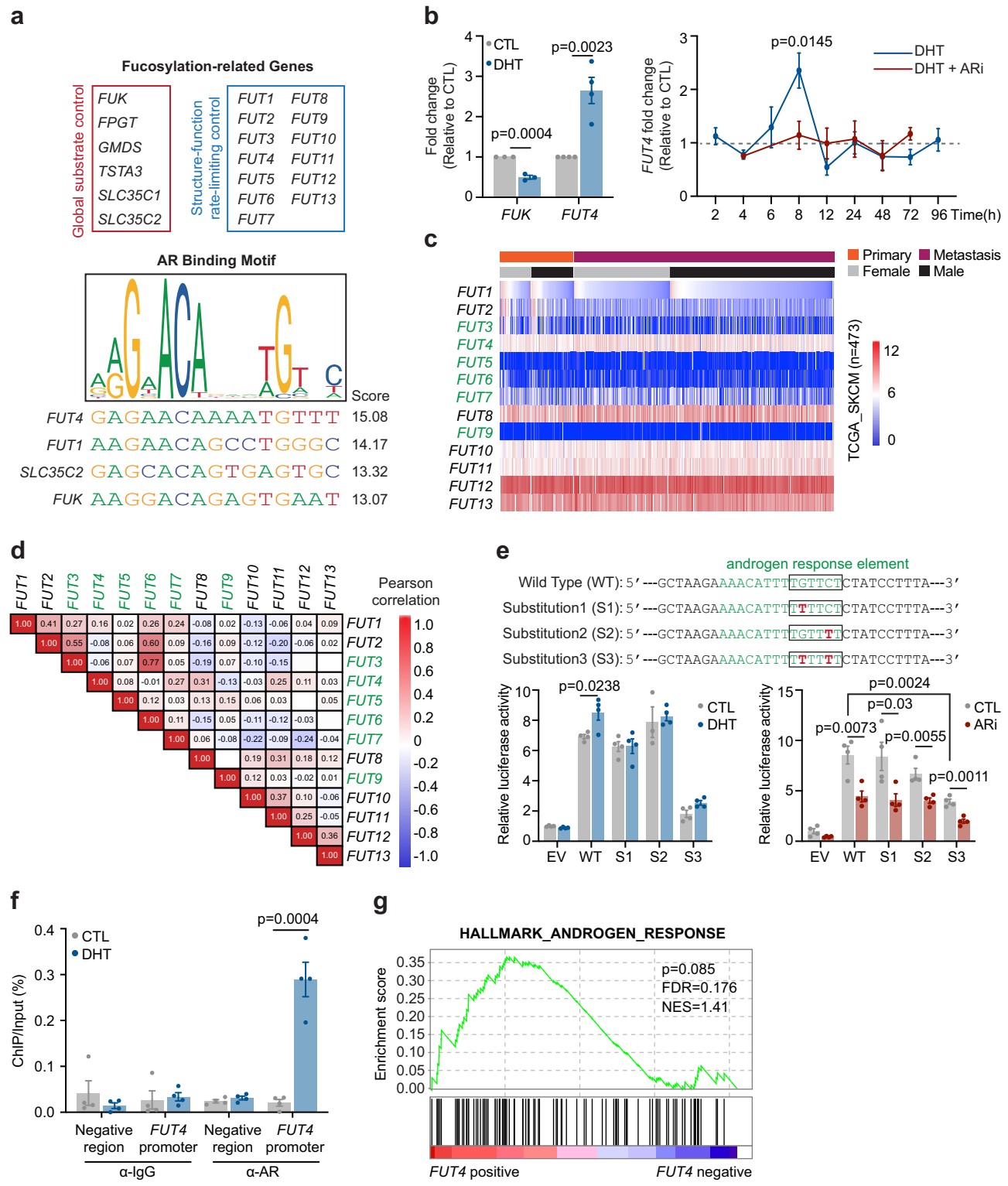

proteins via FUT-specific structural linkages. Due to their differential effects on the stability and activity of target proteins, these distinct linkages (a.k.a., fucosylation subtypes) can result in divergent tumor-suppressive or tumor-promoting effects (Supplementary Fig. 2a, orange dashed box). Previous reports identified FUT4 as a tumorigenic FUT establishing an oncogenic glycoproteome that promotes cancer metastasis[23–25]. Our findings support the concept that androgen/AR reduces global fucosylation levels by transcriptionally downregulating *FUK* while tilting its function toward tumor promotion by upregulating *FUT4* expression. This would help to explain how global fucosylation

levels are reduced in male melanoma patients[16] and how male patients exhibit worse outcomes than female patients. Thus, we focused our subsequent efforts on delineating an AR-FUT4 signaling axis and its contribution to male sex-associated biological effects in melanoma.

## AR transcriptionally upregulates *FUT4* level via binding to an androgen response element (ARE) in the *FUT4* 5′-promoter

In melanoma cells, steroid hormone-depleted CSS medium reduced *FUT4* mRNA levels, which were restored ~3-4-fold either by steroid hormone-replete medium (10% fetal bovine serum (FBS))

**Fig. 2 | AR transcriptionally upregulates *FUT4* expression via binding to the ARE motif in the *FUT4* promoter. a** (*upper*) Nineteen fucose salvage and de novo synthetic pathway genes. (*lower*) JASPAR-predicted AR binding sites in the 5′-promoter of the *FUT4* gene and in the 3′-UTRs of *FUT1*, *SLC35C2*, and *FUK* genes. **b** qRT-PCR analysis of (*left*) *FUK* and *FUT4* mRNA levels in WM793 cells treated ± 100 nM DHT for 8 h (*FUK*, *n* = 3; *FUT4*, *n* = 4 independent experiments), or (*right*), *FUT4* mRNA levels in WM793 cells treated ± 100 nM DHT ± 10 μM AZD3514 (ARi) over 96 h (*n* = 3 independent experiments). **c** Heatmap visualization and **d** correlation matrix for the mRNA levels of 13 *FUTs* in the TCGA_SKCM samples. α-(1,3)-*FUTs* are indicated with green font. For **d**, the numbers represent Pearson correlation coefficients; red or blue color denotes positive or negative correlation, respectively. **e** Luciferase activity of *FUT4* promoter with wild-type (WT) or site-mutant (S1, S2, S3) androgen response element (ARE) relative to empty vector (EV) control in WM793 cells treated ± 100 nM DHT or ± 10 μM ARi for 48 h (*n* = 4 independent experiments). **f** ChIP-qPCR analysis of the enrichment of endogenous AR protein at the −515-502bp 5′-promoter region of the endogenous *FUT4* gene upon DHT treatment for 6.5 h (*n* = 4 independent experiments). **g** Gene set enrichment analysis (GSEA) illustrating the association of *FUT4* expression with Hallmark_Androgen_Response gene signatures in TCGA_SKCM samples. *p*-value is calculated by two-sided permutation test. FDR, false discovery rate; NES, normalized enrichment score. For **b, e, f**, data are presented as mean values ± SEM and *p*-values are calculated by two-sided Student's *t*-test. Source data are provided as a Source Data file.

(Supplementary Fig. 2b) or by supplementation with DHT alone (Fig. 2b, *left*; Supplementary Fig. 2c). Moreover, a single-dose treatment of DHT produced a dynamic induction of *FUT4* expression that acutely peaked at ~8 h, which was abolished by AR inhibitor (ARi) (Fig. 2b, *right*). Consistent with these results, shRNA-mediated knockdown of AR also reduced baseline *FUT4* levels (Supplementary Fig. 2d).

FUT3/4/5/6/7/9 are α-(1,3)-FUTs that fucosylate terminal lactosaminyl glycans yielding Lewis[x] (Le[x]; CD15) and/or sialyl Lewis[x] (sLe[x]; CD15s) structures[26,27]. FUT4 has been reported to predominantly biosynthesize CD15, whereas CD15s epitope has been reported to be predominantly synthesized by FUT3/5/6/7[23,25,26,28,29]. Notably, only *FUT4* is highly expressed in melanoma, and its expression does not appear to correlate with that of other *FUTs* (Fig. 2c, d). Importantly, the regulation of AR is specific to *FUT4*, as no other *FUTs* were altered in expression upon treatment with DHT (Supplementary Fig. 2e). Moreover, the androgen-stimulated expression kinetics of *FUT4* are consistent with previous reports of AR transcriptional activity for well-characterized target genes in prostate cancer (*e.g.*, *PSA*)[30]. However, genes that are classical AR targets in prostate cancer were not significantly altered by androgen stimulation in melanoma cells, suggesting that the dynamic transcriptional repertoire of AR in melanoma is distinct from that in prostate cancer cells (Supplementary Fig. 2f).

To assess direct transcriptional regulation of *FUT4* by AR, we first verified the contribution of a putative androgen response element (ARE) within the *FUT4* 5′-promoter region. To this end, we cloned a 277-bp fragment of the *FUT4* 5′-promoter containing the ARE motif (5′-TGTTCT-3′) into a luciferase reporter construct. Using this wild-type (WT) promoter luciferase construct, we generated mutant promoter constructs in which we individually or doubly mutated C/G to T within the ARE to abolish AR binding[31] (Fig. 2e, *upper*). Whereas DHT and ARi induced and suppressed, respectively, the transcriptional (luciferase) activity driven by the WT promoter, double mutation significantly reduced the promoter activity at the baseline level, and completely abolished all transcriptional activity during DHT treatment, confirming the requirement of this ARE within the *FUT4* promoter for AR-induced transcription (Fig. 2e, *lower*). ChIP-qPCR confirmed direct binding of endogenous AR protein to the endogenous ARE-containing *FUT4* 5′-promoter region following DHT stimulation (Fig. 2f). Together, these data demonstrate that androgen upregulates transcription of *FUT4* by triggering AR binding to the ARE within the *FUT4* 5′-promoter. Notably, our findings are consistent with further gene set enrichment analysis (GSEA) that revealed a strong associative trend between *FUT4* expression and androgen response signatures in a large-cohort melanoma patient expression dataset (TCGA_SKCM) (Fig. 2g).

### Phosphoproteomic profiling identifies cell adhesion signaling as a key downstream effector of the AR-FUT4 axis

To delineate how the AR-FUT4 axis regulates cellular signaling in melanoma, we generated empty vector (EV) control or FUT4-overexpressing (FUT4-OE) melanoma cell lines, which were validated for ectopic expression, specific FUT4 fucosylation activity (using anti-CD15, which recognizes FUT4-synthesized Lewis[x] carbohydrate epitopes[25,26,32,33]), and lack of off-target effect on the expression of other *FUTs* (Supplementary Fig. 3a-c). VIM2, a sLe[x] paralog[34,35], was nearly undetectable in melanoma cells; however, among CD15 and CD15s glycan epitopes, only CD15 was increased in melanoma cells ectopically expressing FUT4 (Supplementary Fig. 3b). Using these cells, we performed global phosphoproteomic profiling on EV and FUT4-OE WM793 and WM1366 cells treated with or without ARi, followed by multi-stage comparative analyses (Fig. 3a). In the first stage of analysis, we identified 368 unique proteins (represented by 484 phosphopeptides) that were reduced ≥2 fold-change in EV-WM793 cells treated with ARi (these phosphopeptides were referred to as "*ARi-reduced phosphopeptides*"). In the second stage of analysis, we dichotomized the *ARi-reduced phosphopeptides* into the following 2 groups: (i) those restored by overexpression of FUT4 (referred to as "*AR-FUT4-dependent effectors*" (95 proteins)), and (ii) those that were not restored by FUT4 overexpression (referred to as "*AR-dependent, FUT4-independent effectors*" (241 proteins)) (Fig. 3a). Using the DAVID platform[36,37], we delineated remarkable functional separation between groups of effectors: whereas AR-dependent, FUT4-independent downstream effectors were enriched in cell division-related pathways, AR-FUT4-dependent effectors were predominantly enriched in cell adhesion-/motility-related pathways (Fig. 3a). Further Ingenuity Pathway Analysis (IPA)[38] was performed on the 141 AR-FUT4-up/downregulated signatures overlapped in WM793 and WM1366 cells (Supplementary Fig. 3d). IPA analysis highlighted adherens junction (AJ) as the highly AR-FUT4-modulated signaling pathway/mechanism (Fig. 3b). To assess if and how AR-FUT4 might alter AJs, we used proximity ligation assay (PLA) to visualize and quantify interactions between N-cadherin, an essential plasma membrane protein that establishes and stabilizes AJ structures, and key cytoplasmic AJ interactors, β-catenin and δ1-catenin (Fig. 3c). Ectopic FUT4 expression reduced the number of N-cadherin:β-catenin and N-cadherin:δ1-catenin interactions per melanoma cell, whereas the depletion of FUT4 or treatment with 2F-peracetyl-fucose (2FF), a pharmacological inhibitor of FUTs[39], enhanced these junction complexes (Fig. 3d, e and Supplementary Fig. 3e−h). The same N-cadherin:catenin interactions were also increased by ARi treatment or AR knockdown but were suppressed by DHT stimulation (Fig. 3f and Supplementary Fig. 3f, i). Together, these data indicate that AR-driven FUT4 signaling alters cell:cell adhesion by disrupting N-cadherin:catenin-containing junctional complexes between melanoma cells (Fig. 3g), supporting the notion that androgen/AR-FUT4 signaling promotes melanoma motility by altering cellular adhesion complexes.

### FUT4 is crucial for androgen/AR-stimulated melanoma migration and invasion in vitro

To validate the phosphoproteomic data and delineate the functional impact of AR-FUT4-AJs signaling on melanoma biology, we performed a series of proliferation and motility assays. Similar to CSS, treatment of WM793 or WM1366 melanoma cells with ARi suppressed their viability and proliferation—an effect that was not rescued by the ectopic expression of FUT4 (Fig. 4a and Supplementary Fig. 4a, b). Further, knockdown of FUT4 did not block DHT-induced proliferation (Fig. 4b). This finding indicates that FUT4 does not mediate

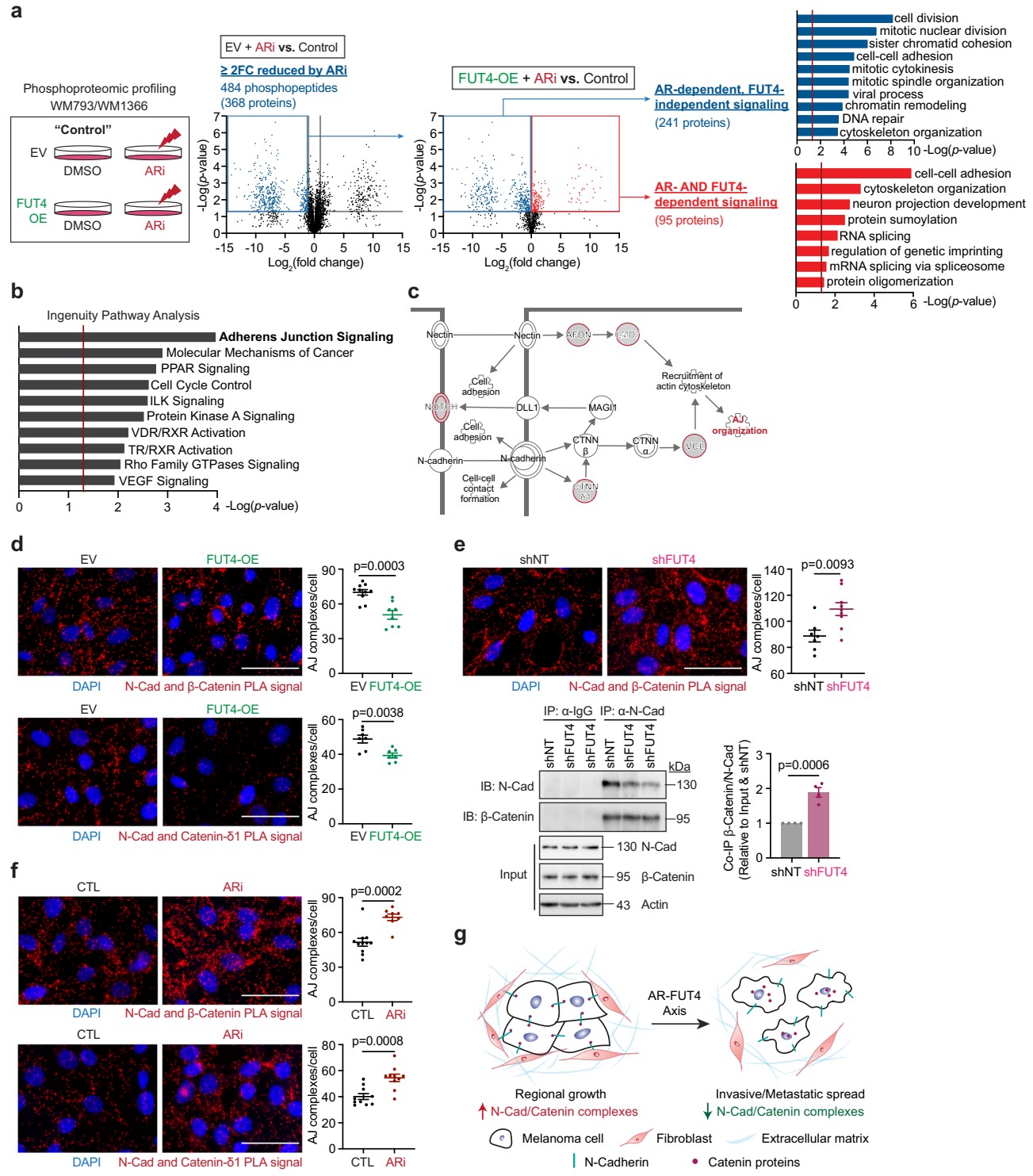

androgen-stimulated melanoma growth and proliferation. Rather, consistent with the observed 2FF-increased and FUT4-OE-decreased AJ formation (Supplementary Fig. 3g, h and Fig. 3d), we found that 2FF and FUT4-OE suppressed and enhanced, respectively, the motility of melanoma cells (Fig. 4c and Supplementary Fig. 4c). Moreover, ectopic FUT4 expression rescued ARi-suppressed invasive capacity of melanoma cells back to baseline levels, whereas depletion of FUT4 abolished DHT-induced cell migration and invasion (Fig. 4d, e and Supplementary Fig. 4d, e). Consistent with the effects of ARi, genetically knockdown of endogenous AR also blunted migration capacity in melanoma cells (Supplementary Fig. 4f).

As FUT8 has been reported to promote motility in melanoma cells[13], we sought to determine if FUT8 might or might not contribute to the motility effects induced by FUT4. To this end, we generated FUT4 overexpressing cells that were knocked down for FUT8 and assessed alterations in motility. FUT8-knockdown alone significantly reduced melanoma cell migration, an effect that was only partially rescued by the simultaneous overexpression of FUT4, suggesting that while both FUTs impact melanoma motility, their functional contributions are not redundant (Fig. 4f and Supplementary Fig. 4g).

Notably, androgen stimulation and FUT4 overexpression in melanoma cells also augmented their matrix (gelatin) degradative capacity

**Fig. 3 | AR regulates melanoma motility/invasion in a FUT4-dependent manner and regulates melanoma proliferation in a FUT4-independent manner. a** (*left*) Global phosphoproteomic profiling of EV vs. FUT4-overexpressing (OE) melanoma cells treated with 10 μM ARi for 48 h. Volcano plots showing phosphopeptides identified by LC-MS/MS to be ≥2-fold-change (FC) reduced by ARi (*center left*) that are or are not rescued by FUT4-OE (*center right*). *p*-values are calculated by two-sided Student's *t*-test. DAVID pathway analysis identifying the enrichment of cell division-related signaling as AR-dependent, FUT4-independent (*upper right*) vs. cell adhesion-related signaling as AR- and FUT4-dependent (*lower right*). In DAVID, one-sided Fisher's Exact test is adopted to measure the gene-enrichment in annotation terms. **b** Ingenuity Pathway Analysis (IPA) highlighting AR-FUT4-regulated protein signatures are predominantly enriched in adherens junction (AJ) signaling. *p*-values are calculated by one-sided Fisher's Exact test. **c** Schematic diagram of AJ signaling between 2 adjacent cells (*adapted from a Qiagen IPA-generated schematic*). Red circles denote hits from our phosphoproteomic profiles. **d** In situ proximity ligation assay (PLA) for N-cadherin and catenin proteins in EV/FUT4-OE WM793 cells. N-cadherin/β-catenin PLA (*upper*): EV, *n* = 10 fields; FUT4-OE, *n* = 8 fields and N-cadherin/δ1-catenin PLA (*lower*): *n* = 7 fields examined over 1 independent experiment. Two other independent experiments in Source Data. Scale bar = 50 μm. **e** (*upper*) PLA and (*lower*) co-immunoprecipitation (co-IP) analyses evaluating the interaction between N-cadherin and β-catenin proteins in shNT/shFUT4 WM793 cells. PLA: shNT, *n* = 7 fields; shFUT4, *n* = 9 fields examined over 1 independent experiment. Two other independent experiments in Source Data. Scale bar=50μm. Co-IP: *n* = 4 biologically independent samples. Uncropped blots in Source Data. **f** PLA staining for N-cadherin and catenin proteins in parental WM793 cells treated ± 10 μM ARi for 48 h. N-cadherin/β-catenin PLA (*upper*): CTL, *n* = 11 fields; ARi, *n* = 8 fields and N-cadherin/δ1-catenin PLA (*lower*): *n* = 10 fields examined over 1 independent experiment. Two other independent experiments in Source Data. Scale bar = 50 μm. **g** Working model of AR-FUT4 axis in disrupting AJs to promote melanoma invasiveness (*the schematic was created using BioRender*). For **d**–**f**, data are presented as mean values ± SEM and *p*-values are calculated by two-sided Student's *t*-test. Source data are provided as a Source Data file.

(Fig. 4g and Supplementary Fig. 4h). In striking support of the notion that FUT4 signaling mediates the pro-invasive but not pro-proliferative effects of androgen/AR in melanoma, whereas treatment of matrigel plug-implanted melanoma spheroids with ARi abrogated both 3D spheroid growth and peripheral invasiveness into the matrigel, the ectopic expression of FUT4 only significantly rescued invasive but not the proliferative capacity of melanoma spheroid under ARi treatment (Fig. 4h). Together, these observations functionally verify our phosphoproteomic analyses highlighting the enrichment of cell division-related pathways as AR-dependent, FUT4-independent as opposed to the enrichment of cell adhesion-/motility-related pathways as AR-/FUT4-dependent signaling in melanoma cells (Fig. 3a).

Melanoma patient dataset analyses support the in vitro findings that AR-FUT4 axis augments the ability of melanoma cells to degrade extracellular matrix (ECM): *FUT4* expression positively correlates with the level of matrix metalloproteinase 2 (*MMP2*) and *MMP9*, gelatinases that are known to degrade ECM and promote tumor metastasis[40] (Supplementary Fig. 4i). Further, *FUT4* levels are significantly higher in metastatic lesions as compared to primary melanoma specimens (TCGA_SKCM and GSE8401 datasets) (Fig. 4i and Supplementary Fig. 4j, k). Moreover, GSEA using the TCGA_SKCM cohort revealed that *FUT4* expression is significantly associated with epithelial-mesenchymal transition (EMT) (*p = 0.006; FDR = 0.018; NES = 1.9*) and melanoma metastasis (*p = 0.018; FDR = 0.018; NES = 1.7*) signatures (Fig. 4j).

Together, these findings demonstrate a crucial role for FUT4 in mediating androgen/AR-stimulated melanoma invasiveness and strongly support the role of the AR-FUT4 axis in promoting melanoma metastasis.

## L1CAM is a key FUT4-fucosylated target that is responsible for AR-FUT4 axis-induced melanoma invasiveness

To delineate the key FUT4-fucosylated target(s) mediating pro-invasive signaling in AR⁺ melanoma cells, we performed comparative proteomic profiling of fucosylated proteins purified from WM793 cells that ectopically expressed FUT4 vs. those that were knocked down for FUT4. Liquid chromatography-MS/MS identified 86 fucosylated proteins that were increased at least 2-fold in FUT4-overexpressing cell lines, whereas 23 fucosylated proteins that were reduced by ≥ 2-fold in FUT4-knockdown cell lines (Fig. 5a, *upper*). The 8 proteins that were found in both of those categories were considered as bona fide FUT4 fucosylation targets (Fig. 5a, *middle*). Of the 8 hits, L1CAM was highlighted by GeneMANIA inter-actome mapping[41] as the most centralized signaling nodal protein, which suggested high probability for significant biological contribution in this context, prompting us to study its potential role as a FUT4 signaling effector (Fig. 5a, *lower*). Fucosylation of L1CAM by FUT8 has previously been reported to stabilize L1CAM at the plasma membrane and facilitate melanoma metastasis[13], but sex-associated regulation and biology of L1CAM has never been reported. Further, our earlier data showing that FUT4 overexpression partially rescues cell motility impaired by FUT8 knockdown suggests distinct functional contributions by each FUT (Fig. 4f and Supplementary Fig. 4g). NetNGlyc[42] and NetOGlyc[43] software predicted 16 N- and 6 O-glycosylation sites in the L1CAM protein (Supplementary Fig. 5a). Lectin pulldown (LPD) assay revealed that AR-knockdown reduced whereas DHT stimulation significantly increased fucosylation of L1CAM (Supplementary Fig. 5b). Similarly, knockdown or over-expression of FUT4 reduced or increased, respectively, the fucosylation of L1CAM (Fig. 5b and Supplementary Fig. 5c). Lectin-mediated PLA (LPLA, a method developed in our laboratory[15,16] that allows for the visualization of specific fucosylated proteins) also confirmed that knockdown or overexpression of FUT4 resulted in reduced or increased, respectively, cellular levels of fucosylated L1CAM (fuco-L1CAM) (Fig. 5c and Supplementary Fig. 5d). Consistent with our data supporting the fucosylation of L1CAM by FUT4, PLA analysis using anti-L1CAM and anti-CD15 showed that ectopically expressed FUT4 induced, whereas AR-knockdown reduced, FUT4-fucosylated L1CAM (Fig. 5d and Supplementary Fig. 5e, f).

Digestion with PNGase F (but not PNGase A), which removes N-linked oligosaccharide chains from glycoproteins[44], increased the electrophoretic mobility of L1CAM and reduced its recognition by immunoblotting antibody following digestion, elucidating that L1CAM is N-glycosylated (*and that the L1CAM antibody exhibits preferential recognition of glycosylated L1CAM*; Supplementary Fig. 5g). Digestion with O-glycosidase, which removes O-linked glycans from glycoproteins[45], resulted in a less pronounced electrophoretic shift and reduction in antibody recognition, suggesting potential but less prominent O-glycosylation of L1CAM (Supplementary Fig. 5g). These findings are consistent with the proportion of predicted N- vs. O-glycans on L1CAM (Supplementary Fig. 5a).

We next sought to delineate the functional contributions of L1CAM to AR-FUT4-regulated melanoma biology by generating melanoma cells that were simultaneously modified for FUT4 (overexpression) and L1CAM (knockdown) (Supplementary Fig. 5h). Whereas manipulation of either or both FUT4 and L1CAM did not impact melanoma viability regardless of ARi treatment (Supplementary Fig. 5i), the depletion of L1CAM abrogated DHT or FUT4-induced melanoma motility (Fig. 5e, f). These observations delineate a central role that fuco-L1CAM plays downstream of the AR-FUT4 axis and its requirement for the invasion-promoting effects of androgen signaling in melanoma.

Together, our findings inform the following working model: in the melanoma, particularly the invasive front of the primary tumor, androgen-activated AR transcriptionally upregulates *FUT4*, which in turn fucosylates L1CAM and disrupts N-cadherin-mediated AJ

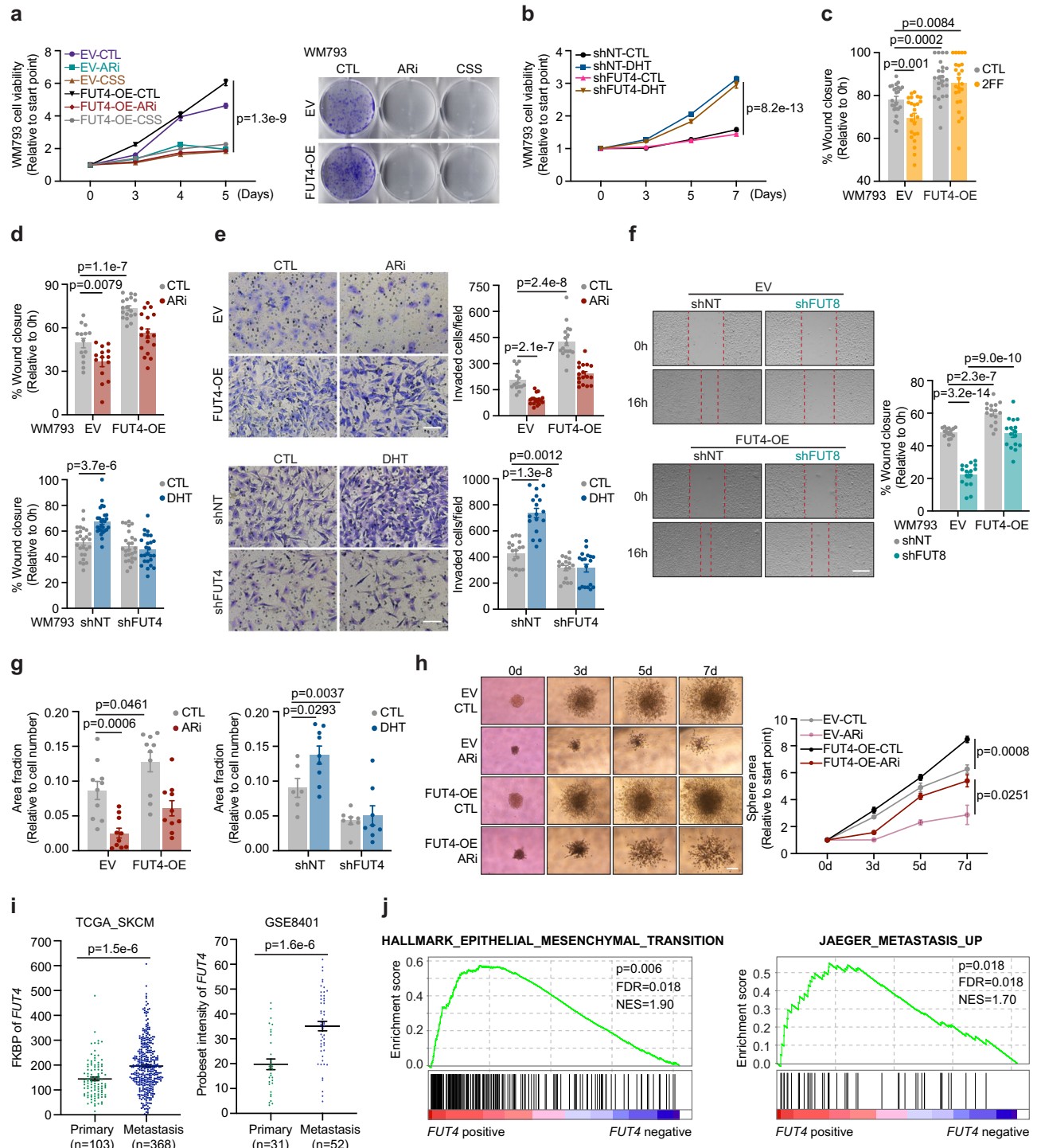

structures that are required for AR-FUT4-enhanced invasiveness and metastatic spread of melanoma cells (Fig. 5g).

## The activation of AR-FUT4-L1CAM-AJ signaling axis in male melanoma tissues

We assessed AR expression and its correlations with downstream effectors in human melanoma specimens by immunofluorescence staining analysis of a 100-case melanoma tissue microarray (TMA). Although total AR exhibit similar levels in tumors from female and male patients, they are substantially higher in metastatic compared with primary tumors of male patients (Supplementary Fig. 6a, b). Moreover, activated AR (nuclear/cytoplasmic AR) is significantly higher in male compared to female tumors (Fig. 6a). Single-cell

segmentation analyses revealed that whereas metastatic lesions contain fewer activated-AR$^{high}$ cells (~20–40% of the tumor cell population) compared with primary tumors (~50–90%), those fewer metastatic cells nonetheless exhibited higher levels of activated AR than primary tumor cells (Fig. 6b). These observations are consistent with a tumor-promoting role for AR, particularly in male melanoma patients. Intriguingly, these data reflect a heterogeneous metastatic landscape comprised of pockets of tumor cells that exhibit higher AR activity than AR$^+$ populations within primary sites. Remarkably, active but not total AR positively correlates with fuco-L1CAM predominantly in stage IIB-III melanomas—pathological staging that is expected to functionally require altered cell adhesion and invasive dynamics to promote subsequent distal metastasis (Fig. 6c and Supplementary

**Fig. 4 | FUT4 is crucial for androgen/AR-stimulated melanoma migration and invasion in vitro. a** (*left*) XTT assay (5 days) and (*right*) clonogenic assay (14 days) of EV/FUT4-OE WM793 cells treated ± 10 μM ARi or cultured in 10% CSS (XTT assay: $n = 6$ biologically independent samples; clonogenic assay: representative images are shown for 3 independent experiments). **b** XTT assay of shNT/shFUT4 WM793 cells treated ± 100 nM DHT for 7 days ($n = 12$ biologically independent samples). **c** Scratch migration assays of EV/FUT4-OE WM793 cells treated ± 250 μM 2F-peracetyl-fucose (2FF) for 3 days ($n = 24$ scratches examined over 3 independent experiments). **d** Scratch migration assay (*upper*: EV-CTL/ARi, $n = 14$ scratches; FUT4-OE-CTL/ARi, $n = 18$ scratches examined over 3 independent experiments. *lower*: shNT-CTL/DHT, $n = 26$ scratches; shFUT4-CTL, $n = 26$ scratches; shFUT4-DHT, $n = 25$ scratches examined over 3 independent experiments) and **e** Matrigel invasion assay (*upper*: EV-CTL, $n = 15$ fields; EV-ARi, $n = 17$ fields; FUT4-OE-CTL/ARi, $n = 17$ fields examined over 2 independent experiments. *lower*: shNT-CTL, $n = 19$ fields; shNT-DHT, $n = 17$ fields; shFUT4-CTL, $n = 17$ fields; shFUT4-DHT, $n = 18$ fields examined over 3 independent experiments) of EV/FUT4-OE WM793 cells treated ± 10 μM ARi for 48 h (*upper*) or shNT/shFUT4 WM793 cells treated ± 100 nM DHT for

48 h (*lower*). Scale bar = 200 μm. **f** Scratch migration assay of FUT4/FUT8 double-modified WM793 cells. $n = 16$ scratches examined over 3 independent experiments. Scale bar = 400 μm. **g** FITC-gelatin degradation assay of EV/FUT4-OE WM793 cells treated ± 10 μM ARi for 48 h (*left*; $n = 10$ fields examined over 3 independent experiments) or shNT/shFUT4 WM793 cells treated ± 100 nM DHT for 48 h (*right*; shNT-CTL, $n = 6$ fields; shNT-DHT, $n = 9$ fields; shFUT4-CTL/DHT, $n = 8$ fields examined over 3 independent experiments). **h** 3D spheroid cell invasion assay with EV/FUT4-OE WM793 cells treated ± 10 μM ARi for 7 days (EV-CTL, $n = 7$; EV-ARi, $n = 3$; FUT4-OE-CTL, $n = 4$; FUT4-OE-ARi, $n = 4$ biologically independent samples). Scale bar = 200μm. **i** Comparison of *FUT4* mRNA levels between primary vs. metastatic melanomas in TCGA_SKCM and GSE8401 datasets. **j** GSEA illustrating the association of *FUT4* expression with Hallmark_Epithelial_Mesenchymal_Transition and Jaeger_Metastasis_Up gene signatures in TCGA_SKCM samples. *p*-values are calculated by two-sided permutation test. FDR, false discovery rate. NES, normalized enrichment score. For **a–i**, data presented as mean values ± SEM and *p*-values are calculated by two-sided Student's *t*-test. Source data are provided as a Source Data file.

Fig. 6c, d). Indeed, N-cadherin/β-catenin AJ complexes exhibit a negative correlation with activated AR only in male melanomas (Fig. 6d), supporting the notion that AR signaling promotes melanoma invasive and metastatic capacity by undermining N-cadherin-mediated AJ structures (Fig. 3g). Together, these findings support the role of AR in driving aggressive melanoma pathogenesis via fucosylation of L1CAM and abrogation of AJ complexes in male patients.

### The AR-FUT4 axis promotes lung intravascular melanoma colonies in vivo

To delineate the functional contribution of AR-FUT4 signaling in driving melanoma metastasis in vivo, we implanted EV or FUT4-OE WM793 melanoma cells into immunodeficient male NSG mice. Mice were fed with control (CTL) or enzalutamide (Enzal) diet once primary tumors were palpable (~ 50–100 mm³) (Fig. 7a). Enzalutamide significantly decelerated the growth of primary EV tumors (~ 50%; Fig. 7b). Intriguingly, although WM793 cells were initially derived from a primary melanoma and were not previously reported to exhibit significant spontaneous metastatic ability in nude mice[46,47], we observed induction of what appeared to be intravascularly trapped "micrometastatic" WM793 colonies in the lungs of mice bearing ectopic FUT4-expressing tumors, which were partially reduced by enzalutamide (FUT4-OE-CTL diet: ~13–14 intravascular colonies/lung; FUT4-OE-Enzal diet: ~6-7 intravascular colonies/lung; Fig. 7c and Supplementary Fig. 7a, b). Due to the limited metastatic capacity of WM793 cells (EV-CTL diet: ~2 intravascular colonies/lung), enzalutamide did not elicit a significant inhibitory effect at the baseline level (EV-Enzal diet: ~3–4 intravascular colonies/lung) (Fig. 7c). These findings verify our in vitro observations and confirm that AR-induced FUT4 significantly promotes invasive and metastatic capacity in melanoma (Fig. 4).

### Discussion

Increasing studies support tumor-promoting roles for androgen in melanoma proliferation, motility, and refractoriness to therapeutic agents[4–6]. Indeed, our phosphoproteomic analyses show that androgen induces global signaling pathway changes in melanoma, particularly in those governing cell division and motility/invasiveness. Intriguingly, our data indicate that the latter is controlled predominantly by fucosylation. Here, we report an intersection between AR-regulated transcription and protein fucosylation that plays a crucial mechanistic role in driving the androgen-stimulated metastatic capacity of melanoma. Specifically, our data show that androgen-activated AR directly transcriptionally upregulates the expression of the tumorigenic *FUT4*, which fucosylates a key mediator of metastasis, L1CAM. Together, our preclinical and patient data demonstrate that activation of the AR-FUT4 axis contributes to the disparately worse

clinical outcomes observed in male patients by disrupting cellular adhesion and driving tumor cell migration and invasion.

Sex disparities are clinically evident in regard to incidence and mortality rates across multiple non-reproductive organ cancers including bladder, kidney, colorectal, liver, esophagus, head and neck, brain, skin, and blood cancers[48]. Generally, male patients exhibit increased susceptibility and unfavorable prognosis compared with female patients for those cancers, which is likely attributable to a complex interplay of sex-related discrepancies in occupational/behavioral traits, genetics, epigenetics, sex hormones, and immune responses[49]. Emerging studies are unraveling immunologic mechanisms underlying the sexual dimorphism in cancers. Sex-specific behaviors of myeloid-derived suppressor cells (MDSCs) and T cells have been found to drive sex-associated pathological differences in glioblastoma, where male patients display increased monocytic MDSCs and exhausted CD8⁺ T cells in the tumor microenvironment, contributing to accelerated tumor progression but enhanced response to single-agent anti-PD-1 treatment in preclinical models[50,51]. Mechanistically, sex chromosome-derived intrinsic regulation (*e.g.*, via X-chromosome inactivation (XCI) escape genes) potentially underpins the sex differences in T-cell exhaustion[51]. Moreover, mutation and differential expression of Y-chromosome genes have also been reported to play critical roles in altering T cell-mediated anti-tumor effects in bladder and colon cancers[52,53]. Additionally, androgen-driven T-cell exhaustion and dysfunction have recently been observed prominently in the models of melanoma, prostate cancer, bladder cancer, and colon cancer[54–56]. Notably, androgen signaling blockade can effectively rescue effector T cell function and potentiate immune checkpoint blockade efficacy in these studies[54–56].

Androgen receptor signaling has been well established as a key regulator of carcinogenesis, progression, and the development of therapeutic resistance in prostate cancer[57–60]. However, its emerging associations with cellular functions and particularly with sexual dimorphisms observed in the clinic have piqued increasing research interest in the context of other tumor types that have not classically been considered as sex hormone-associated. Melanoma has historically exhibited prominent sex-related differences, where higher circulating testosterone levels correlate with increased melanoma risk[61]. In a recent clinical study, Vellano et al. reported that AR signaling impairs the efficacy of BRAF/MEK-targeted therapy, providing compelling implications of AR blockade as a potential strategy for improving therapeutic responses in melanoma[6]. However, our understanding of AR-regulated cellular and molecular mechanisms in controlling melanoma biology— knowledge that is crucial for informing how AR signaling can be leveraged for patient stratification or therapeutic targeting—has been relatively limited. A recent study from Ma et al. reported that AR signaling can trigger melanomagenesis by activating DNA repair processes in

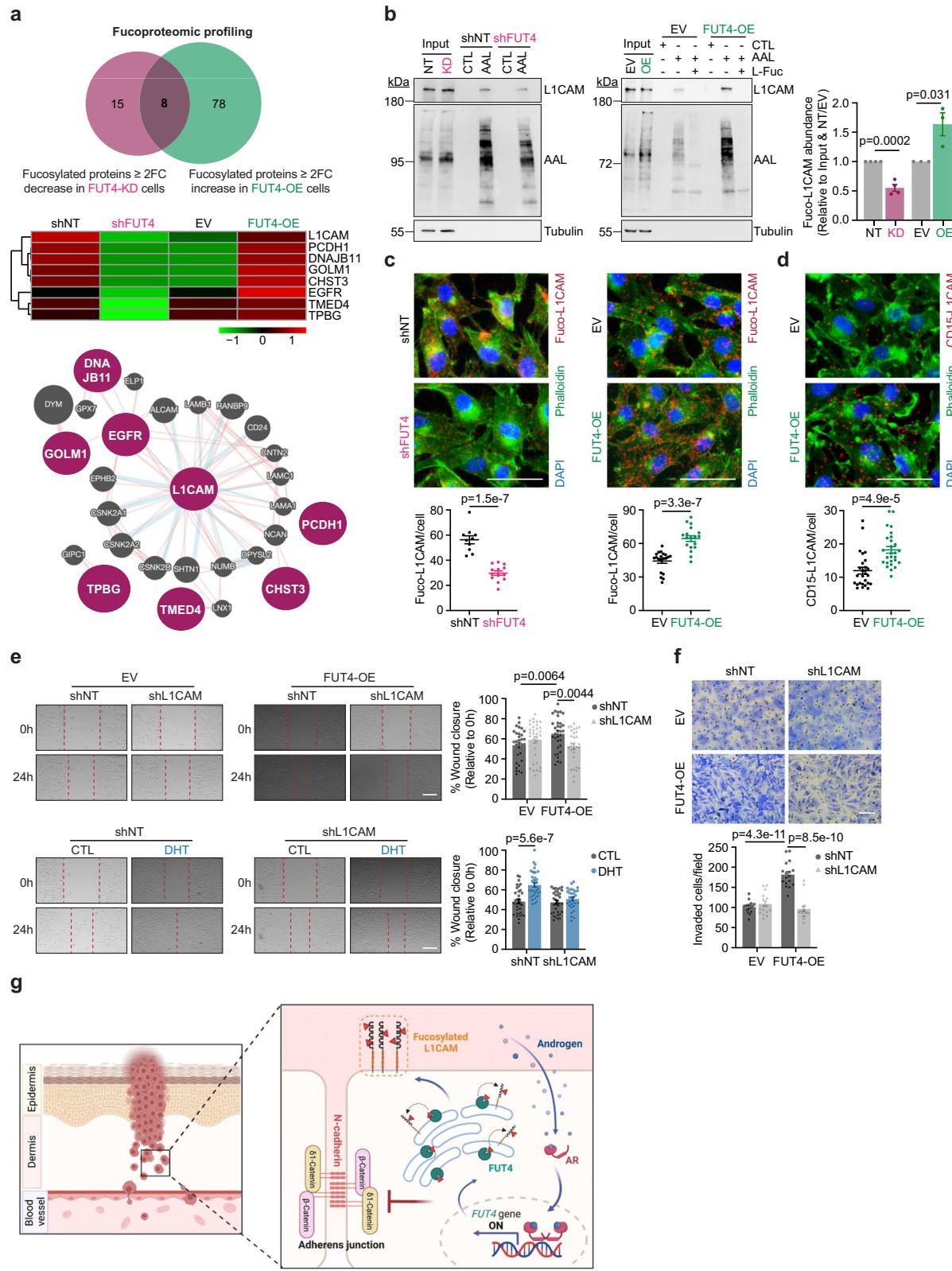

melanoma cells, which further maintains malignant growth potential[5]. This is consistent with our cell-based functional and global phospho-proteomic analyses showing that androgen stimulates cell division and DNA repair signaling pathways in an AR-dependent, FUT4-independent manner (Fig. 3a). Regarding melanoma motility, AR was previously reported to promote melanoma metastasis via transcriptional upregulation of miRNA-539-3p, which impacts MITF and AXL signaling[4].

However, if androgen is necessary for activating this AR-regulated axis, as well as the pathophysiological context to which this axis functionally contributes, remain unclear. Our work, together with these previous studies highlight how AR signaling can potently stimulate melanoma tumorigenesis, growth, distal metastasis, and therapeutic resistance.

Our study identifies a significant and direct downstream mechanistic intersection between the non-canonical transcriptional

**Fig. 5 | FUT4-fucosylated L1CAM is required for AR-FUT4-induced melanoma invasiveness. a** (*upper and middle*) Fucoproteomic profiling of shNT/shFUT4 (FUT4-knockdown, FUT4-KD) and EV/FUT4-OE WM793 cells identified 8 proteins that are specifically fucosylated by FUT4 and (*lower*) GeneMANIA interactome mapping of the 8 protein hits highlighting L1CAM as the most centralized signaling protein (*adapted from schematic generated in GeneMANIA*). **b** Lectin pulldown (LPD) followed by IB analysis of L1CAM protein in shNT/shFUT4 (*left*) and EV/FUT4-OE (*center*) WM793 cells. Column chart (*right*) shows densitometric quantification for the blots (*n* = 3 independent experiments). Uncropped blots in Source Data. **c** Lectin-mediated proximity ligation assay (LPLA) staining for fucosylated-L1CAM (fuco-L1CAM) in shNT/shFUT4 WM793 cells (*left*; shNT, *n* = 10 fields; shFUT4, *n* = 13 fields examined over 3 independent experiments) and EV/FUT4-OE WM793 cells (*right*; EV, *n* = 20 fields; FUT4-OE, *n* = 18 fields examined over 4 independent experiments). Scale bar = 50 μm. **d** PLA staining for CD15 and L1CAM proteins in EV/FUT4-OE WM793 cells (*n* = 27 fields examined over 3 independent experiments).

Scale bars = 50 μm. **e** Scratch migration assays of FUT4/L1CAM double-modified WM793 cells (*upper*; EV-shNT, *n* = 31 scratches; EV-shL1CAM, *n* = 32 scratches; FUT4-OE-shNT, *n* = 33 scratches; FUT4-OE-shL1CAM, *n* = 30 scratches examined over 3 independent experiments) and shNT/shL1CAM WM793 cells treated ± 100 nM DHT for 48 h (*lower*; EV-shNT/shL1CAM, *n* = 36 scratches; FUT4-OE-shNT, *n* = 34 scratches; FUT4-OE-shL1CAM, *n* = 31 scratches examined over 3 independent experiments). Scale bar, 400 μm. **f** Matrigel invasion assay on FUT4/L1CAM double-modified WM793 cells (EV-shNT, *n* = 17 fields; EV-shL1CAM, *n* = 16 fields; FUT4-OE-shNT/shL1CAM, *n* = 17 fields examined over 2 independent experiments). Scale bar = 200 μm. **g** Working model: During melanoma invasion through the dermis, DHT-activated AR transcriptionally upregulates *FUT4*, which then triggers melanoma migration and invasion through increased fuco-L1CAM and impaired junction structures (*the schematic was created using BioRender*). For **b**–**f**, data are presented as mean values ± SEM and *p*-values are calculated by two-sided Student's *t*-test. Source data are provided as a Source Data file.

repertoire of AR and oncogenic protein fucosylation that facilitates melanoma invasiveness potentially during both early-stage establishment and late-stage metastatic progression in androgen-responsive melanomas. FUT4 is well-known as a tumorigenic FUT, the expression of which is associated with the induction of cell proliferation[62,63], EMT/invasiveness[64], and multi-drug resistance[65–67] in a variety of cancers (*e.g.*, breast cancer, lung cancer, melanoma, hepatocellular carcinoma, and colorectal cancer). Particularly, in melanoma, increased FUT4 was previously reported to be correlated with high metastatic potential and invasive phenotype[24,68]. However, the upstream regulator as well as the downstream fucosylated targets of FUT4 are poorly characterized and have never been associated with sex in cancer. Interestingly, androgen/AR has previously been linked to transcriptional regulation of *Fut1, Fut2, Fut4 and Fut9* in normal male mouse reproductive tissues, although the biological significance and functional contributions are unknown[69,70]. Therefore, our study delineates for the first time the pathological transcriptional regulation of FUT4 by AR and their downstream effectors and functional contributions to the metastatic spread of human cancer (melanoma).

Our global proteomic analyses identified cell-cell adhesion proteins, particularly those involved in adherens junctions, as predominantly modulated by the AR-FUT4 axis. The subtype shift from E-cadherin to N-cadherin is a critical early event in melanoma progression[71]. N-cadherin-mediated AJs are dynamically altered to achieve regional migration and distant invasion of melanoma cells: although increased AJs enhance the ability of transformed melanocytes to migrate through the basement membrane into the dermis, reduction in N-cadherin-mediated AJs is required at the late stage to promote tumor escape from the primary site[72–75]. As the AR⁺ cell lines (WM793 and WM1366) used in our study were derived from vertical growth phase (VGP) melanoma lesions which have invaded into the dermis, they exhibit baseline loss of E-cadherin, and conversely, a significant number of N-cadherin-dominant AJs. Most importantly, the activation of the AR-FUT4 axis was found to disrupt these junctional complexes to augment melanoma metastasis in our study. Moreover, we identified fucosylated-L1CAM as a key downstream effector of the AR-FUT4 axis. Core fucosylation of L1CAM via FUT8 was previously reported to impair L1CAM cleavage, stabilizing it on the cell surface and enhancing the ability of L1CAM to support melanoma invasion[13]. Here, we revealed that AR-FUT4-mediated terminal fucosylation of L1CAM also confers metastatic behavior to melanoma cells. Our data provide a sex-associated functional connection between AJs and fucosylated-L1CAM downstream of AR-FUT4 signaling. Although L1CAM was reported to stimulate invasion in breast cancer by disrupting AJ structures[76], the precise underlying molecular mechanism is unclear. In melanoma, their exact modes of action—whether parallel, intersecting, or successive—to mediate invasiveness remain to be determined.

While our study has identified a previously undisclosed AR-downstream tumorigenic signaling axis, our understanding of the

upstream regulation (and deregulation) of AR activity in melanoma—whether it is classically and predominantly androgen-stimulated—remains unclear. Non-canonical upstream activation of AR in melanoma is an important consideration for future studies of androgen- and AR-regulated melanomagenesis, as independent of its canonical ligand DHT, AR is known to be phosphorylated and activated by other factors (*e.g.*, AKT, HER2, and ACK1 kinases) in prostate cancers[77–80]. In this regard, consistent with previous studies on prostate cancer and melanoma[5,81], we also observed a stronger inhibitory effect on melanoma by AZD3514 compared to enzalutamide. This is important, as AZD3514 blocks both ligand-dependent and ligand-independent activities of AR[81], whereas enzalutamide inhibits only ligand-dependent AR[82]. Our findings suggest that in melanoma cells, stimulation of AR activity at least partially bypasses androgen. Further studies are needed to elucidate the upstream activation of AR. In addition, nonclassical androgen signaling through ZIP9 has also been reported to impact tumorigenesis, particularly in AR-null cancer cells[83,84]. However, specific contributions of androgen-ZIP9 signaling across different cancers remain unclear: in breast and prostate cancer cells, androgen activates inhibitory G protein (Gi) through ZIP9 and induces apoptosis[84,85], whereas in melanoma, androgen-ZIP9 signaling through YAP1 was reported to induce proliferation[86]. Future studies are expected to elucidate the relative functional contribution of these signaling mechanisms to AR⁺ vs. AR⁻ melanomas under different stage- and therapy-specific contexts.

Currently, clinical trials are underway investigating therapeutic strategies for combining AR blockade with chemotherapy, targeted therapy, or immune checkpoint blockade in patients with advanced cancers (NCT04926181, NCT01974765, NCT03207529, NCT02684227, NCT02312557). Combining the administration of AR antagonists with BRAF/MEK inhibitors has been recognized as a potential strategy for enhancing the responsiveness of melanomas to the inhibitors[6]. However, considering the severe side effects of AR inhibition, it will be crucial to develop effective stratification methods for melanoma patients. Our study has identified, in addition to systemic levels of testosterone, tumor-specific AR and its fucosylation signaling effectors (FUT4, fucosylated-L1CAM, and AJs), as potential risk factors for disease progression and as promising prognostic biomarkers. Along these lines, the inhibition of AR downstream effectors (*e.g.*, FUT4 and L1CAM) may represent more effective targets for suppressing AR-driven tumorigenesis with fewer unintended toxicities than general AR inhibition. Additional studies are needed to evaluate if this AR-regulated signaling is conserved in driving other cancer types. Glycobiological studies on fucosylated-L1CAM are needed to map the fucosylation sites, dissect their effects on L1CAM functions, and assess their contributions to androgen-driven melanoma metastasis. Together, our findings identify and delineate an androgen-/AR-regulated signaling axis that drives melanoma malignancy, reinforcing melanoma as a sex-associated cancer and highlighting new therapeutic opportunities for the clinical management of melanoma invasiveness.

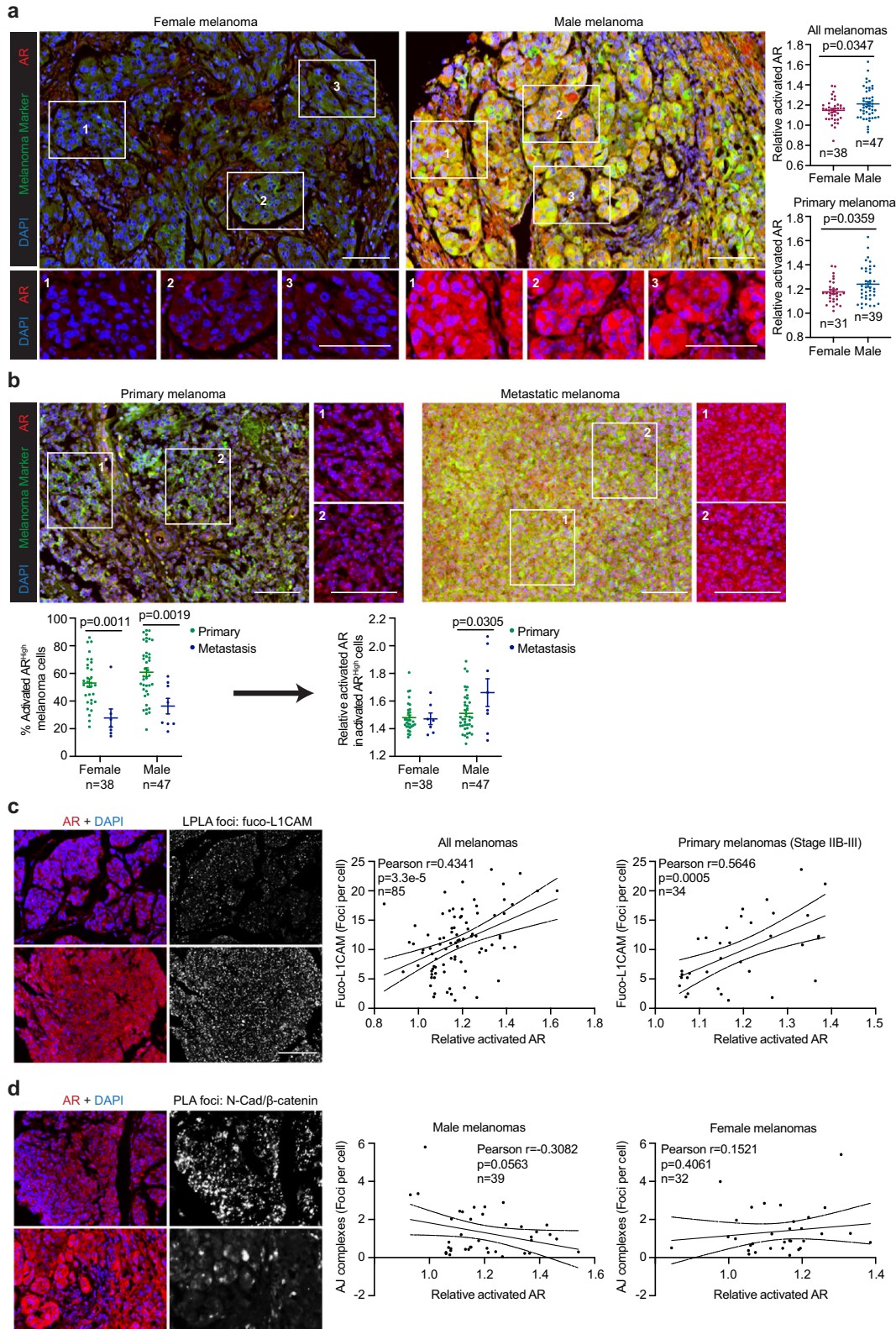

**Fig. 6 | The activation of AR-FUT4-regulated signaling in male melanoma tissues. a** Representative images (*left*) and quantification (*right*) of multiplexed IF staining for tumor-specific activated AR in female vs. male melanoma tissues. Relative activated AR = the ratio of nuclear/cytoplasmic AR. **b** Representative images (*upper*) and quantification (*lower*) comparing activated AR levels between primary and metastatic melanomas in female and male patients (high level: above median level among all melanoma cells across the whole TMA). Representative images (*left*) and correlation analyses (*right*) of **c** activated AR and fuco-L1CAM (LPLA foci), as well as **d** activated AR and N-cadherin/β-catenin junction complexes (PLA foci). For **a, b**, data are presented as mean values ± SEM and *p*-values are calculated by two-sided Student's *t*-test. For **c, d**, *p*-values are determined by two-sided correlation test based on Pearson's coefficient. All scale bars = 100 μm. Melanoma marker: a cocktail of MART-1 + Tyrosinase + gp100. Source data are provided as a Source Data file.

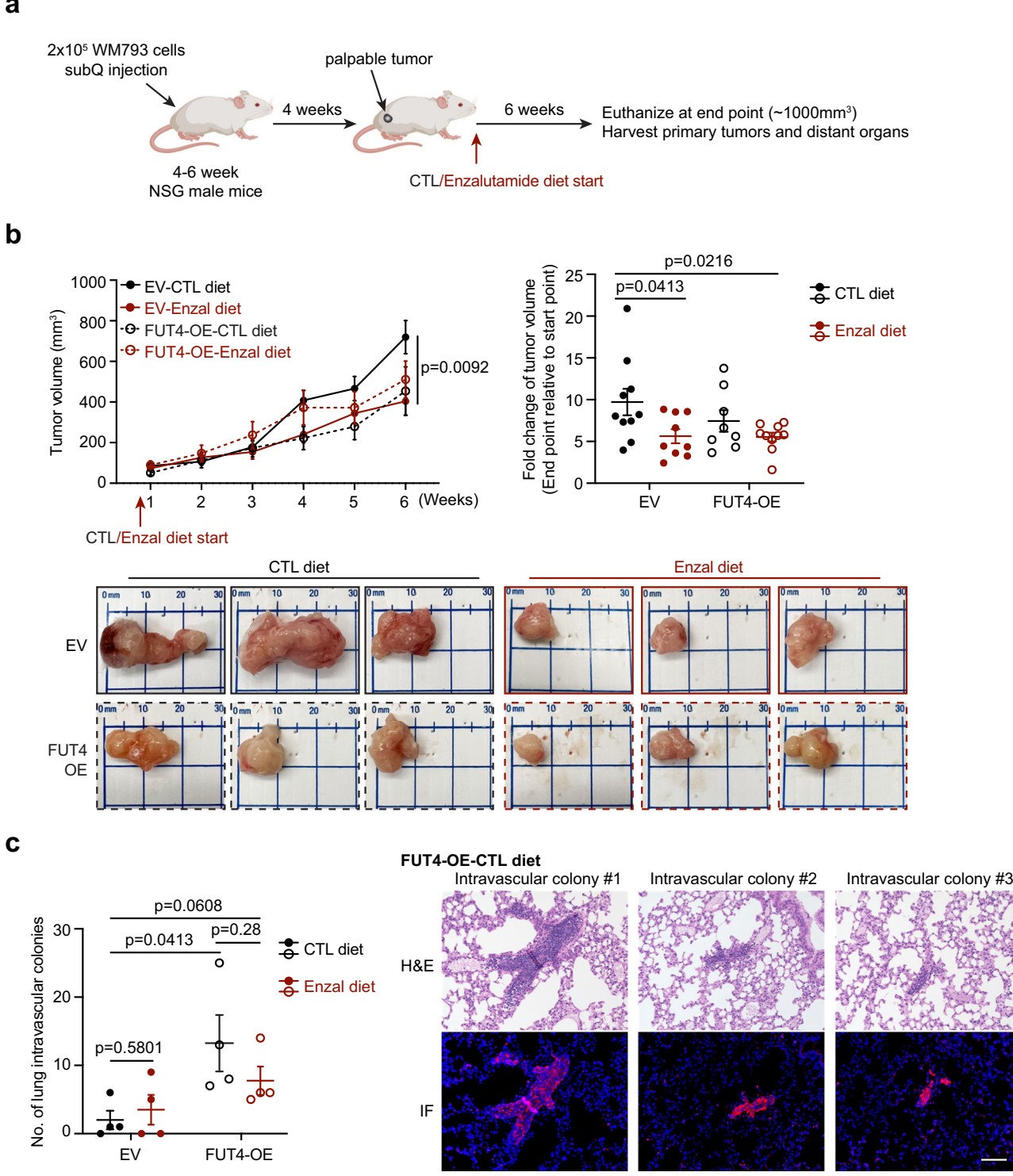

**Fig. 7 | The AR-FUT4 axis promotes accumulation of lung intravascular melanoma colonies in vivo. a** Experimental design for mouse tumor model. **b** (*upper*) The growth curve and end-point fold-change of EV/FUT4-OE WM793 tumors subcutaneously implanted in NSG mice fed with control (CTL) or enzalutamide (Enzal) diet (EV-CTL diet, *n* = 10 mice; EV-Enzal diet, *n* = 9 mice; FUT4-OE-CTL diet, *n* = 8 mice; FUT4-OE-Enzal diet, *n* = 10 mice). (*lower*) Representative images of primary tumors at the end point. **c** (*left*) Quantification of lung intravascular melanoma colonies (*n* = 4 lungs per group) and (*right*) representative H&E staining and corresponding IF staining of lung intravascular melanoma colonies in mice harboring FUT4-OE melanoma tumors fed with CTL diet. Scale bar = 100 μm. For **b, c**, data are presented as mean values ± SEM and *p*-values are calculated by two-sided Student's *t*-test. Source data are provided as a Source Data file.

## Methods

Our research complies with all relevant ethical regulations. All animal experiments were performed in accordance with an Institutional Animal Care and Use Committee protocol (IACUC protocol, #IS00010075) approved by the University of South Florida. The commercial human melanoma tissue microarrays were purchased from US Biomax (#ME1004h), with an ethics statement, "All tissue is collected under the highest ethical standards with the donor being informed completely and with their consent. We make sure we follow standard medical care and protect the donors' privacy. All human tissues are collected under HIPPA approved protocols. All animal tissues are collected under IACUC protocol. All samples have been tested negative for HIV and Hepatitis B or their counterparts in animals and approved for commercial product development".

### Cell lines, cell culture, and cell treatments

The following cell lines (and their respective catalog numbers) were from American Type Culture Collection (ATCC; Manassas, VA): A375 (#CRL-1619) and HEK293T (#CRL-3216). The following cell lines (and their respective catalog numbers) were from Rockland Immunochemicals (Limerick, PA): WM983A (WM983A-01-0001); WM983B (WM983B-01-0001); WM1366 (WM1366-01-0001); WM115 (WM115-01-0001); WM266-4 (WM266-4-01-0001); WM164 (WM164-01-0001); WM793 (WM793-01-0001); and LU1205 (1205LU-01-0001). The SM1 murine melanoma cell line was obtained from the Smalley laboratory; SKMEL19 and Meljuso cell lines were obtained from the Karreth laboratory; the IPC298 cell line was obtained from the Tsai laboratory, and the LNCaP human prostate cancer cell line was obtained from the Wang laboratory at Moffitt Cancer Center & Research Institute. All cell lines were STR footprinted, and their identities were verified by the Moffitt Molecular Genomics Core. Melanoma and HEK293T cell lines were cultured in Dulbecco's modified Eagle's medium (DMEM) (Cytiva) supplemented with 10% bovine serum (BS; PEAK Serum). LNCaP cells were cultured in RPMI-1640 medium (Corning) with 10% fetal bovine serum (FBS; PEAK Serum). Cells were maintained at 37 °C in a humidified 5% carbon dioxide ($CO_2$) incubator. All cells in this study were confirmed to be mycoplasma-free before experiments using the PlasmoTest Mycoplasma Detection Kit (InvivoGen).

For in vitro experiments requiring DHT treatment, the cells were first steroid hormone-depleted by culture in phenol red-free DMEM (Gibco) supplemented with 10% charcoal-stripped serum (CSS; Gibco) for 48 h, followed by treatment with 100 nM DHT (Sigma-Aldrich) or vehicle control (methanol) in fresh media. In general, 1–100 nM of DHT is widely applied in cell-based studies of prostate cancer which are highly AR-expressed and androgen responsive[87–91]. We chose 100 nM because (i) we hypothesized that melanoma cells would need more DHT in order to signal through their lower levels of expressed AR, and (ii), our preliminary experiments showed that 100 nM DHT resulted in consistent changes in fucosylation, whereas lower doses were less consistent (*data not shown*). Other melanoma studies have also previously used 20–100 nM DHT for in vitro cell-based studies[5,86]. For AR inhibitor (ARi) treatment, melanoma cells were treated with 10 µM AZD3514 (Adooq Biosciences) or vehicle control (DMSO). For 2F-peracetyl-fucose (2FF) treatment, cells were treated with 250 µM 2FF (Sigma-Aldrich) or vehicle control (DMSO) for 72 h. Cells were harvested at indicated time points following treatment.

### Establishment of stably expressing cell lines

The coding sequence of the *FUT4* gene (NCBI RefSeq; CCDS8301.1) in melanoma cells was cloned into the pLenti-C-Myc-DDK-IRES-Puro lentiviral gene expression vector (OriGene) between AscI and XhoI restriction sites. Lentiviral particles were generated using HEK293T cells transfected with control (empty pLenti-C-Myc-DDK-IRES-Puro) or pLenti-C-Myc-DDK-IRES-Puro-FUT4 lentiviral vectors along with VSVG and Δ8.9 packaging vectors as previously described[14].

WM793 and WM1366 melanoma cells were subsequently infected with lentivirus, followed by antibiotic selection (1.5 µg/ml puromycin (InvivoGen)). Using the same lentiviral production and infection methods, shRNA-encoding plasmids (pLKO.1 lentiviral vector) (MISSION shRNA, Sigma-Aldrich) were used to stably knock down human *FUT4* and *L1CAM* genes in melanoma cells. Three shRNAs for each gene were used for lentiviral infection, after expression validation, the one with the most knockdown effect was utilized for functional assays.

### Site-directed mutagenesis and secreted luciferase reporter assay

The 277-bp wild-type *FUT4* promoter region (SwitchGear Genomics) containing the putative AR-binding element (5'-AAACATTTTGTTCTC-3') was cloned from WM793 cDNA into the pMCS-Gaussia Luc vector (ThermoFisher Scientific) between the SacI and BamHI restriction enzyme sites. ARE mutant promoter constructs (S1, S2, and S3) were generated using the Q5 Site-Directed Mutagenesis Kit (NEB) per the manufacturer's protocols. Sequence verification was performed by Eton Bioscience. Primers were synthesized by IDT and listed in Supplementary Data 1.

Melanoma cells were co-transfected with EV or wild-type/mutant FUT4 promoter-Gaussia luciferase constructs and pCMV-Cypridina (constitutive control) vector (ThermoFisher Scientific) at a ratio of 50:1 using Lipofectamine 3000 (Invitrogen). At 24 h after transfection, cells were treated either with methanol/DHT or with DMSO/ARi. At 48 h after treatment, samples of media were collected and subject to Pierce Gaussia/Cypridina Luciferase Flash Assay (ThermoFisher Scientific) per the manufacturer's protocols. Luciferase activity was measured using a Promega GloMax Luminometer. Gaussia luciferase values were normalized to Cypridina values to control for the transfection efficiency.

### Subcellular fractionation

The Nuclear Extract Kit (Active Motif) was used for isolating nuclear and cytoplasmic proteins from the indicated melanoma cells per the manufacturer's protocols. Fractions were stored at −80 °C until ready to use.

### Immunoblot

Cells were washed in cold PBS and lysed in RIPA Buffer containing protease inhibitor and phosphatase inhibitor tablets (ThermoFisher Scientific). After sonication, protein concentrations were determined using DC Protein Assay (Bio-Rad). Equal amounts of heat-denatured proteins were loaded onto 8–12% SDS-PAGE gel, and then the separated proteins on the gel were transferred onto a polyvinylidene fluoride (PVDF) membrane (Bio-Rad). Membranes were blocked either in 5% non-fat milk in TBST or in Carbo-Free blocking solution (Vector Laboratories) in TBST for 30 min, followed by incubation with primary antibodies overnight at 4 °C. The next day, membranes were washed and incubated with horseradish peroxidase (HRP)- or infrared dye (IRDye)-conjugated secondary antibodies for 1 h at room temperature (RT). After 3–5 washes in TBST, membranes were developed by film or imaged on an Odyssey Fc Imaging System (LI-COR Biosciences). All antibodies used for immunoblotting in this study are commercially available and detailed information are provided in Reporting Summary.

### qRT-PCR

RNA was extracted from indicated cells using GenElute Mammalian Total RNA Miniprep Kit (Sigma-Aldrich) per the manufacturer's protocols. cDNA was synthesized using the High-Capacity cDNA Reverse Transcription Kit (Applied Biosystems). Subsequent qRT-PCR was performed using PerfeCTa SYBR Green SuperMix (Quantabio) on a CFX96 Real-Time PCR Detection System (Bio-Rad). Each reaction system was composed of 1 µl cDNA, 5 µl SYBR Green, 0.5 µl 10 µM forward and reverse primer mix, and 3.5 µl DNase/RNase-Free $H_2O$ (Invitrogen).

Samples were amplified under the following cycling conditions: 95 °C for 10 min; 40 cycles of 95 °C for 15 s, 55 °C for 60 s, and 72 °C for 30 s; and 95 °C for 15 s. Gene expressions were calculated as fold changes through the formula: $2^{-\Delta\Delta CT}$. Primer sequences were obtained from PrimerBank and synthesized by IDT (Supplementary Data 1).

## Immunofluorescent staining

Indicated coverslip-grown cells were fixed in the fixation buffer (4% formaldehyde + 2% sucrose in PBS) for 20 min at RT. Fixed cells were washed with PBS, followed by permeabilization with 0.4% Triton X-100 + 0.4% IgG-free bovine serum albumin (BSA) for 20 min at RT. Cells were then blocked with 2% IgG-free BSA for 1 h at RT. Primary antibodies (listed in Reporting Summary) were diluted in 1% BSA and applied on cells overnight at 4 °C. The next day, after washing 3-5 times in the washing buffer (0.2% Triton X-100 + 0.2% IgG-free BSA in PBS), cells were incubated with Alexa Fluor-conjugated secondary antibody (Invitrogen) diluted in 1% BSA for 2–3 h in the dark at RT. Coverslips were washed 3–5 times with washing buffer prior to mounting on glass slides using Vectashield Antifade Mounting Media (Vector Laboratories). Images were acquired using a Keyence BZ-X710 fluorescence microscope and analyzed using FIJI software (NIH).

## ChIP-qPCR

ChIP assay was performed as described previously[14,92,93]. Briefly, cells were crosslinked using 1% formaldehyde for 10 min at RT and then quenched with 0.125 M glycine for 5 min at RT. Cells were lysed and sonicated to shear the DNA to ~500-bp fragments. After pre-clearing with A/G beads (Sigma-Aldrich) for 1–2 h, 200 µg of lysate was subject to immunoprecipitation using 1.5 µg of control IgG or AR antibody overnight at 4 °C. 30 µl of blocked A/G beads were added to the chromatin-antibody solution for incubation for 5 h at 4 °C. After washing, the AR-DNA complexes were eluted from the beads and reversed from cross-linking. After treating with 0.2 mg/ml RNase A (Invitrogen) and 0.4 mg/ml Proteinase K (NEB), DNA was then purified with Purelink PCR Purification Kit (Invitrogen) and subjected to standard qRT-PCR analysis. The PCR primers for the ChIP assays are listed in Supplementary Data 1.

## Dual-luciferase reporter assay

Indicated cells cultured in 24-well plates were co-transfected with pGL3-EV/pGL3-ARR2-Firefly constructs and pCMV-Renilla control vectors at a 10:1 ratio using Lipofectamine 3000 (Invitrogen). At 24 h after transfection, cells were treated with methanal or DHT. At 24 h after treatment, the cells were washed with PBS and lysed. Firefly and Renilla luciferase activities were measured using the Dual-Luciferase Reporter Assay System Kit on a Promega GloMax Luminometer per the manufacturer's protocols. Firefly luciferase values were normalized to Renilla luciferase values to control for transfection efficiency.

## In vitro functional assays

**XTT assay.** Melanoma cells were plated (1000 cells/well) and treated as indicated in 96-well plates. Cell viability/proliferation was measured at the indicated time after treatment by adding 50 µl/well of 1 mg/ml XTT reagent (Invitrogen) and 5 mM PMS (Phenazine methosulfate, Sigma-Aldrich), to final concentrations of 0.2 mg/ml and 6.25 µM, respectively. After incubation at 37 °C for 4 h, absorbances at 450 nm were measured using an iMark microplate absorbance reader (Bio-Rad).

**BrdU incorporation assay.** Coverslip-grown DHT-treated melanoma cells were incubated with a final concentration of 20 µM 5-bromo-2′-deoxyuridine (Abcam) at 37 °C for 30 min. Labeled cells were fixed, permeabilized, and subjected to the following IF staining procedure. 1:70 dilution of DNAse I (Sigma-Aldrich) was applied on cells for 45 min at 37 °C, followed by washing and incubating cells with anti-BrdU antibody (Sigma-Aldrich) overnight at 4 °C. The cells were next washed

5 times and incubated with Alexa Fluor-conjugated secondary antibody (Invitrogen) at RT for 1–3 h in the dark. After 3-5 additional washes, the coverslips were mounted on glass slides with Vectashield Antifade Mounting Media (Vector Laboratories).

**Clonogenicity assay.** Melanoma cells were seeded in 6-well plates at a density of 5,000 cells/well. Cells were treated as described. Fresh medium ± drug was replaced every three days. After 2 weeks of culture, colonies were fixed with freezer-cold (−20 °C) 100% methanol and stained with 0.5% w/v crystal violet in 20% ethanol.

**Scratch assay.** Melanoma cells were seeded in 12-well plates and grown to confluence. Cells were cultured in fresh media ± drug for 24 h prior to scratch. The drug-containing media from each well was removed and centrifuged to remove floating cells. Each well was washed with PBS once, and a 0.5–1 mm wide scratch was made through the entire center of the well using a sterile pipette tip. After washing with PBS twice, the centrifuged media was added back into each respective well. The cells were immediately imaged after scratch as a starting point and were imaged every 4–8 h for 48 h using a standard light microscope.

**Transwell invasion assay.** Transwell assays were performed in Matrigel Invasion Chambers (24-well, 8 µm pore size, Corning) per the manufacturer's protocols. Briefly, 750 µl of DMEM containing 10% FBS or 10% CSS ± indicated treatments were placed in the bottom well as a chemoattractant, and 500 µl of melanoma cell suspensions (including $5 \times 10^4$ cells) in serum-free DMEM ± indicated treatments were seeded onto the top matrigel-coated chambers. After 16–48 h incubation, the non-invaded cells were removed from the upper surface of the chamber membrane with cotton swabs. The invaded cells on the lower surface of the membrane were fixed with 100% methanol and stained with 0.5% crystal violet in 20% ethanol. Six-eight fields/membrane were photographed under a standard light microscope (Leica DMi1).

**Gelatin degradation assay.** Melanoma cells were pretreated with DHT or ARi for 24 h before seeding into 8-well chamber slides coated with FITC-conjugated gelatin (Invitrogen) ($3 \times 10^4$ cells/well). After 24 h of culture with indicated treatments, the cells were fixed, permeabilized, and blocked as described above in the IF staining section. The cells were subsequently stained with AlexaFluor 594-Phalloidin (Thermo-Fisher Scientific). The slides were mounted and imaged using a Keyence BZ-X710 fluorescent microscope. Quantification of the areas devoid of FITC (i.e., degraded gelatin areas) was performed on Fiji software (NIH).

**3D spheroid invasion assay.** Melanoma cells were cultured in 25-µl suspension droplets of medium ± ARi (400 cells/droplet) in the hanging droplet culture plates (Nunc). After 4 days of culture, aggregated cell spheroids were transferred into 96-well plates coated with 75 µl 0.7% solidified soft agarose. Next, 75 µl of liquefied Matrigel was overlaid on top of the spheroids and was allowed to polymerize at 37 °C for 20 min. Next, 100 µl of warm culture medium ± 10 µM ARi was added to each well. Images of the spheroids were captured at 0, 3, 5, and 7 days after initial transfer using a light microscope (Leica DMi1). Subsequent analysis was performed using Fiji software (NIH).

**Deglycosylation assay.** 20 µg of melanoma cell lysates were digested with PNGase A, PNGase F, or O-glycosidase (NEB) under denaturing reaction conditions per manufacturer's instructions.

## Proximity ligation assay (PLA) and lectin-mediated proximity ligation assay (LPLA)

PLA and LPLA were performed with Duolink In Situ PLA Kits (Sigma-Aldrich) per the manufacturer's protocols. Briefly, coverslip-grown

melanoma cells were treated as indicated, followed by fixation, permeabilization, and blocking as described above in the IF staining section. Next, 2 primary antibodies (for PLA) or 1 primary antibody + 1 biotinylated lectin (for LPLA) diluted in Duolink antibody diluent were added onto coverslips and incubated overnight in a humidity chamber at 4 °C. Cells were subsequently washed in PBST (PBS + 0.05% Tween-20) for 3 x 5 min and incubated with Alexa Fluor 488-phalloidin (ThermoFisher Scientific) and goat anti-biotin (For LPLA) (Vector Laboratories) for 2 h at RT. The coverslips were then washed with PBST for 3 x 5 min and incubated with the PLA probes for 1 h at 37 °C. This was followed by 2 x 5 min washes with buffer A, 30 min ligation reaction at 37 °C, two more 5-min washes with buffer A, and final amplification reaction for 100 min at 37 °C in the dark. Cells were washed in buffer B for 2 x 10 min and in 0.01 x buffer B for 1 min and then were mounted on the glass slides. Images were acquired with a Keyence BZ-X710 fluorescence microscope. The number of nuclei and PLA punctae were quantified using the Fiji software (NIH).

### Lectin pull-down (LPD)

Control agarose beads (ThermoFisher Scientific) and AAL lectin-conjugated agarose beads (Vector Laboratories) were pre-blocked with 2% IgG-Free BSA on a rotator at 4 °C for 2–3 h. Cells were lysed with the IP buffer (1% Triton X-100; 20 mM Tris-HCl, pH 7.4; 150 mM NaCl in ddH2O + protease and phosphatase inhibitors). After sonication, the concentration of lysates was determined by DC Protein Assay (Bio-Rad). For each pull-down reaction, 400 µg of lysates were diluted 1:4 with the dilution buffer (detergent-free IP buffer) and rotated with 30 µl of pre-blocked beads overnight at 4 °C. The next day, the beads were washed 3 times with PBS. Beads-bound proteins were eluted and denatured by boiling in 20 µl of 2x Laemmli buffer at 95–100 °C for 10 min. The resulting proteins in the supernatant were subjected to further IB analysis. For the "L-fucose washing" group, control and AAL agarose beads were precleared and blocked with 500 mM L-fucose solution. After pulldown, beads were washed with 500 mM L-fucose solution.

### Co-immunoprecipitation

Cell lysates were precleared with 50% protein A/G agarose beads (Sigma-Aldrich) for 1–2 h at 4 °C with gentle agitation. Beads were pelleted and discarded by centrifuging at 14,000 g for 5 min at 4 °C. Next, 400 µg of precleared lysates was incubated with 1.5 µg of anti-N-cadherin or anti-IgG antibodies in 500 µl dilution buffer overnight on a rotator at 4 °C. Subsequently, 30 µl of pre-blocked A/G beads was added to the lysate-antibody mixture and rotated for 5 h at 4 °C. The beads were washed 3 times with PBS, resuspended in 20 µl 2x Laemmli buffer, and boiled for 10 min at 95–100 °C. The resulting samples were further subjected to SDS-PAGE and immunoblotting with the indicated antibodies. Antibody details are provided in Reporting Summary.

### Phosphoproteomic analyses: sample preparation, LC-MS/MS, and data analysis

EV and FUT4-OE melanoma cells were treated with DMSO or ARi for 48 h as indicated. The 48-h period is a timepoint with maximal AR activation and downstream changes in FUT4-induced fucosylation, that also coincided with significant alterations in biological phenotypes. This long timepoint assessment also better reflects the physiological in vivo context, where the tumors would be constantly exposed to DHT. The cells were then lysed with urea lysis buffer, and protein concentrations were determined by DC Protein Assay (Bio-Rad). Next, 1 mg of proteins was reduced with 4.5 mM dithiothreitol (DTT), alkylated with 10 mM iodoacetamide (IAA), and trypsin digested. The tryptic peptide solution was desalted using reverse-phase Sep-Pak C18 cartridges (Waters). Following lyophilization, phosphopeptides (pSTY) were enriched using PTMScan Phospho-Enrichment IMAC Fe-NTA

Magnetic Beads (Cell Signaling Technology) on a KingFisher Flex Purification System (ThermoFisher Scientific). Subsequently, the enriched peptides were vacuum concentrated using a speed vac and resuspended in 15 µL of solvent A (2 % acetonitrile (ACN) and 0.1% formic acid (FA)).

A nanoflow ultra-high-performance liquid chromatography (RSLC, Dionex) connected to a Q-Exactive Plus mass spectrometer (Thermo-Fisher Scientific) was used for tandem mass spectrometry peptide sequencing. The LC-MS/MS was performed by the Proteomics Core at Moffitt Cancer Center. Briefly, peptide mixtures were first loaded onto a pre-column (2 cm × 100 µm ID packed with C18 reversed-phase resin, 5 µm, 100 Å) and washed for 8 min with aqueous solvent A. The trapped peptides were eluted onto the analytical column (C18, 75 µm ID x 25 cm, 2 µm, 100 Å, Dionex). The 120-minute gradient was programmed as follows: 95% solvent A for 8 min, solvent B (90% ACN + 0.1% FA) from 5% to 38.5% in 90 min, then solvent B from 50 to 90% in 7 min and held at 90% for 5 min, followed by solvent B from 90% to 5% in 1 min and re-equilibrate for 10 min. The flow rate on the analytical column was 300 nl/min. Twenty tandem mass spectra were collected using data-dependent acquisition (DDA) following each survey scan. The resolution settings were 60,000 and 17,500 for MS1 and MS/MS, respectively. The isolation window was 2.0Th with an offset of 0.5.

The data were processed and analyzed using MaxQuant software (version 1.5.2.8.)[94]. The fragment mass tolerance was set to 20 ppm. Peptides with a minimum of 7 amino acids and a maximum of 2 missed cleavages were considered. Methionine oxidation, N-terminal acetylation, and serine/threonine/tyrosine phosphorylation were selected as variable modifications. Carbamidomethylation of cysteine was used as the fixed modification. The false discovery rate (FDR) was applied at 0.05. MaxQuant data was further normalized with IRON (Iterative Rank-Order Normalization) within each dataset (Supplementary Data 2). Pathway enrichment analyses were performed on DAVID (Functional Annotation Tool) and QIAGEN IPA platforms.

### Fucoproteomic analyses: sample preparation, LC-MS/MS, and data analysis

EV/FUT4-OE and shNT/shFUT4 melanoma cells were lysed, and 800 µg of lysates were used for lectin pull-down as described above. Control and AAL agarose beads were washed with PBS and subjected to a short stack SDS-PAGE followed by in-gel digestion as follows: Briefly, proteins bound to beads were denatured at 95 °C for 5 min and subjected to SDS-PAGE. After 20 min of electrophoresis, the gel was rinsed with water and stained with InstantBlue solution (Abcam) for 30 min. Gel bands were excised and minced, followed by de-staining with 50 mM Ambic/50% methanol, reduction with 25 mM Ambic/2 mM TCEP, alkylation with 25 mM Ambic/20 mM IAA, and trypsin digestion. Peptides were extracted from the gel by incubation in 50% acetonitrile/ 0.1% TFA for 20 min at RT. The resulting peptide solutions were purified by the Ziptip procedure (Millipore). The eluted peptides were dried by speed vac and suspended in 15 µL of solvent A. LC-MS/MS and data analyses were performed as described above (Supplementary Data 2).

### TCGA and microarray dataset analysis

Gene expression data in 473 skin cutaneous melanoma cases from The Cancer Genome Atlas were downloaded from UCSC Xena Functional Genomics Explorer[95]. The GSE8401 microarray dataset, including 83 melanoma cases, was downloaded from NCBI-Gene Expression Omnibus (GEO)[96,97] Patients were stratified based on sex and disease stages. The co-expression of FUT4 and MMP2/MMP9 in the TCGA_SKCM dataset was analyzed on cBioPortal[98,99].

### Gene set enrichment analysis

Global mRNA expression profiles of the TCGA_SKCM dataset were downloaded from Broad GDAC Firehose and were subjected to GSEA

to evaluate the association of *FUT4* level with hallmark gene sets and melanoma metastasis gene signatures (JAEGER_METASTASIS_UP). For GSEA, the expression of *FUT4* was applied as the phenotype label, and "No_Collapse" was used for the gene symbol. The metric for ranking genes was selected as 'Pearson'. Analyses were performed using GSEA software (version 4.2.3)[100,101].

## Melanoma TMA staining, Image acquisition, processing, and analysis

Paraffin-embedded tissue slides were dewaxed by heating at 70 °C for 30 min prior to deparaffinization and rehydration through xylene and ethanol series washes. Antigen retrieval was conducted by heating the slides in citrate buffer (10 mM citric acid, 0.05% Tween 20, pH 6.0) using a pressure cooker. Tissues were blocked using Duolink blocking solution (Sigma-Aldrich) for 30 min at 37 °C. Primary antibodies were applied on tissue slides overnight at 4 °C. The next day, PLA and LPLA reactions were carried out as detailed above. Tissues were then incubated with conjugated primary antibodies for 4–5 h at RT. Following five washes in PBS, the slides were mounted with Vectashield Antifade Mounting Media (Vector Laboratories).

The multiplex fluorescence TMA images were scanned and quantitatively analyzed by the Analytic Microscopy Core at Moffitt Cancer Center. Briefly, immunofluorescently stained ME1004h TMA slides were imaged with a Zeiss Imager Z2 microscope and Zen software version 2.3 (Carl Zeiss AG, Germany) using a 20x/0.8NA objective lens and Hamamatsu Flash 4.0 V3 CMOS camera (Hamamatsu, Japan). An X-Cite Xylis broad spectrum LED light source (Excellitas Technologies Corp., Canada) and DAPI, FITC, DSRED, and CY5 filter cubes were used to excite and capture emissions of each fluorophore. Whole slide images (4 z-planes at 0.75 μm intervals) were automatically captured using the tile scan mode with parabolic surface saddle autofocus on anchor points placed on each TMA tissue spot. The resulting images were stitched, background corrected using Zen software, and background reference imaged for each fluorescence channel. Finally, maximum projection images of the stitched Z-stacks were created for image analysis.

The maximum projection TMA images (CZI format) were imported into Definiens Tissue Studio version 4.7 (Definiens AG, Germany) where individual cores were identified using the software's automated TMA segmentation tool. Each core was manually segmented into Tumor and Non-Tumor regions using the Region of Interest (ROI) tool within the software and AF488 melanoma marker and an H&E slide image as a guide. Using these selected ROIs, a nucleus segmentation algorithm was applied to the DAPI channel to identify nuclei and a cell growth algorithm was used to create individual cell boundaries. A minimum size threshold of 15 μm² was used to refine the nucleus and cell segmentations. Next, the SPOT detection tool was used to identify PLA foci using consistent intensity and size threshold settings across all TMA cores. Finally, data for each TMA core was extracted to excel file format, including mean fluorescence intensity (MFI) values for the AF568 channel and the number of PLA/LPLA foci for the cell, cytoplasm, and nucleus compartments for both the Tumor and Non-Tumor regions.

## Mouse models

Four-to-six-week-old male C57BL6 mice were purchased from Charles Rivers Laboratories and male NSG mice were obtained from the Jackson Laboratory. All mice were housed in the Vincent A. Stabile Research Building animal facility at the Moffitt Cancer Center, in rooms on a standard 12-h–12-h light cycle, with a temperature range of 68–72 °F and humidity range of 30–70%. Mice were randomly divided into groups. In general, 10 mice per indicated cohort were used to accommodate for incidental loss. $1 \times 10^6$ SM1 or $2 \times 10^5$ WM793 melanoma cells were subcutaneously injected into the right flanks of C57BL6 mice or NSG mice, respectively. Physical castration of C57BL6

mice was performed at 1.5 weeks prior to SM1 cell implantation. When WM793 tumors were established and became palpable (~4 weeks post-implantation), NSG mice were randomized to the control or enzalutamide diet group. Enzalutamide chows were formulated by Research Diets, Inc. As described before[102,103], enzalutamide (ChemieTek) was mixed with ground mouse chow (Teklad 2018) at 300 mg/kg chow. Control diet was the same ground chow but without enzalutamide. The diet was irradiated and stored at 4 °C before use. Based on the daily food intake, approximately 40 mg/kg body weight of the drug was delivered per day to study mice. Primary tumors were measured with a digital caliper every week and volumes were calculated using the formula: (length x width x depth)/2. At the endpoint (tumor volume ~1.5 cm³; the maximal tumor volume allowed by our Institutional Review Board (that was never exceeded)), mice were euthanized, and lungs were resected and fixed in formalin. Tissue embedding in paraffin, sectioning, and hematoxylin and eosin (H&E) staining were performed by Tissue Core at Moffitt Cancer Center. The quantification of lung metastasis was performed using a Keyence BZ-X710 microscope. Immunofluorescent staining of adjacent unstained lung sections with human melanoma markers (S100 + MART-1) was further conducted for metastasis validation.

## Statistical analysis and reproducibility

The data are presented as mean ± standard error of the mean (SEM). All western blots were performed at least twice. All in vitro experiments are representative of at least 3 independent replicates. The detailed replicate information is provided in the Figure Legends and Source Data file. For mouse model, 10 mice per group were utilized to achieve the ability to detect a 10% difference in tumor development between any 2 conditions with a *p*-value of 0.05 and a power of 0.80, and a 20% change with a *p*-value of 0.05 and a power of 0.95. In our experience, 10 mice per group have been more than sufficient to provide statistical power and buffer for incidental loss of mice due to factors outside of our control (*e.g.*, unexpected death, tumor ulceration/did not graft successfully). For in vitro experiments, the number of samples/conditions analyzed, and the detailed statistical methods applied are described in the Figure Legends. Sample size was not predetermined statistically but was chosen based on previous experience and published literatures to ensure adequate statistical power and obtain statistically relevant results. Comparisons between 2 groups were calculated with unpaired two-sided Student's *t*-test unless otherwise mentioned. The correlations were calculated based on Pearson's coefficient. Graphs and statistical tests were generated with Prism 9 (GraphPad) unless otherwise indicated. *p*-value < 0.05 was considered as statistically significant. For all experiments, samples/cells/mice were randomly allocated into control and treatment groups. Proteomics data were collected blindly by the Proteomics and Metabolomics Core at Moffitt Cancer Center & Research Institute. No blinding was performed for the other experiments considering the complexity of the study as well as the unbiased data collection/analysis/statistical tests performed with certified software. All software used for data collection and data analysis are provided in Reporting Summary.

## Reporting summary

Further information on research design is available in the Nature Portfolio Reporting Summary linked to this article.

# Data availability

The mass spectrometry proteomics data have been deposited to the ProteomeXchange Consortium via the PRIDE[104] partner repository with the dataset identifiers PXD047335 (Phosphoproteomics) and PXD047337 (Fucoproteomics). Initial matrixes of proteomics data after MaxQuant analysis and IRON or BBSR normalization are provided in Supplementary Data 2. The main data supporting the findings of this study are available within the article and its Supplementary Information

files. In-house script based on dplyr and tidyverse R packages were used in data frame analysis. The TCGA_SKCM dataset was downloaded from UCSC Xena Functional Genomics Explorer and Broad GDAC Firehose. The GSE8401 microarray dataset was downloaded from NCBI-Gene Expression Omnibus (GEO). All antibodies/software used in this study are listed in the Reporting Summary. All primer sequences/special chemicals and reagents/commercial kits/plasmids applied in this study are listed in Supplementary Data 1. The source data are provided with this paper. Additional data are available from the corresponding author upon reasonable request. Source data are provided with this paper.

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

## Acknowledgements

We are grateful to all Lau laboratory members at Moffitt Cancer Center for critical readings of this manuscript. We would like to acknowledge Ms. Melissa Meister for administrative support. This work has been supported in part by the Moffitt/USF Vivarium, Flow Cytometry Core, Biostatistics, Analytic Microscopy, Tissue Core Facilities (we would like to thank in particular Mr. Noel Clark and Ms. Jodi Balasi for technical support) and Advanced Analytical and Digital Laboratory at H. Lee Moffitt Cancer Center & Research Institute, an NCI designated Comprehensive Cancer Center (P30-CA076292). Support from NCI grant R01CA241559 (to E.L.) and a Florida Department of Health Bankhead Coley Grant 22B02 (to E.L.) are gratefully acknowledged.

## Author contributions

Conceptualization: E.L., Q.L.; Methodologies: E.L., Q.L., E.A., D.K.L., B.F., A.M-M., V.I., K.M.G.; Computational Analysis: Q.L., J.J., Y.T.; Resources: J.M., M.G.W., J.A.W., J.Q.; Writing: E.L., Q.L.; Supervision: E.L., J.M.K.; Funding Acquisition: E.L. All of the authors commented on the manuscript.

## Competing interests

The authors declare no competing interests.
