## [Peer Review File · Nature Communications]

REVIEWERS' COMMENTS

Reviewer #1 (Remarks to the Author):

I thank the authors for their additional experiments and their replies to my comments. In my opinion the additional work has substantially improved the manuscript and I have no further comments. I would only like to suggest to add "figure 3 for editor and reviewers" to the extended data figures, as it nicely demonstrates the glycan phenotype of the FUT4-overexpressing melanoma cells.

Reviewer #3 (Remarks to the Author):

In this manuscript (NCOMMS-23-34684-T), Liu et al. report that sex hormone regulated fucosylation mechanistically contributes to sex differences in melanoma. Sex differences in cancer are emerging as a key consideration with studies showing sex differences in genetic, epigenetic, and immune responses in many non-reproductive cancers, which likely underlie the sex difference observed in cancer incidence and outcome. Here, the authors provide strong support for a new regulatory axis in melanoma whereby androgen receptor upregulates fucosyltransferase 4 (FUT4) expression, which results in the fucosylation of L1 cell adhesion molecule (L1CAM), interfering with adherens junctions. These findings are exciting as they show how sex hormones, in this case androgen, can impact tumor cell phenotypes and overall cancer growth. The studies are done in a rigorous manner, utilize in vitro/in vivo studies and human sample validation. The manuscript is well written and the conclusions made are supported by the data provided. I have no major studies that need to be done prior to publication.

While my enthusiasm is very high for this work, I would recommend an edit to the narrative. The manuscript starts off with the idea of sex differences in cancer and this concept should be re-visited in the discussion. An additional paragraph on sex differences in cancer covering recent papers and putting this exciting work into context would be useful. These studies can include work from the Moran (PMID 35322234), Li (PMIDs 35420889, 37344596), DePinho (PMID 37344599), and Lathia (PMIDs 37378557, 32300059) laboratories.

REVIEWERS' COMMENTS

Reviewer #1 (Remarks to the Author):

I thank the authors for their additional experiments and their replies to my comments. In my opinion the additional work has substantially improved the manuscript and I have no further comments. I would only like to suggest to add "figure 3 for editor and reviewers" to the extended data figures, as it nicely demonstrates the glycan phenotype of the FUT4-overexpressing melanoma cells.

We thank the reviewer for the appreciation of our study and constructive comments which have helped us to strengthen and improve our manuscript. We agree with the reviewer and now added the "figure 3 for editor and reviewers" to the Supplementary Figure 3b to elucidate the FUT4-mediated glycan epitopes in melanoma cells. This new Supplementary Figure panel is referenced in the revised main text lines 176-177. We have also added a few lines of text in the preceding subsection to preface/introduce the relevant FUT-mediated structures (revised Main Text, lines 146-148).

Reviewer #3 (Remarks to the Author):

In this manuscript (NCOMMS-23-34684-T), Liu et al. report that sex hormone regulated fucosylation mechanistically contributes to sex differences in melanoma. Sex differences in cancer are emerging as a key consideration with studies showing sex differences in genetic, epigenetic, and immune responses in many non-reproductive cancers, which likely underlie the sex difference observed in cancer incidence and outcome. Here, the authors provide strong support for a new regulatory axis in melanoma whereby androgen receptor upregulates fucosyltransferase 4 (FUT4) expression, which results in the fucosylation of L1 cell adhesion molecule (L1CAM), interfering with adherens junctions. These findings are exciting as they show how sex hormones, in this case androgen, can impact tumor cell phenotypes and overall cancer growth. The studies are done in a rigorous manner, utilize in vitro/in vivo studies and human sample validation. The manuscript is well written and the conclusions made are supported by the data provided. I have no major studies that need to be done prior to publication.

While my enthusiasm is very high for this work, I would recommend an edit to the narrative. The manuscript starts off with the idea of sex differences in cancer and this concept should be re-visited in the discussion. An additional paragraph on sex differences in cancer covering recent papers and putting this exciting work into context would be useful. These studies can include work from the Moran (PMID 35322234), Li (PMIDs 35420889, 37344596), DePinho (PMID 37344599), and Lathia (PMIDs 37378557, 32300059) laboratories.

We thank the reviewer for the enthusiasm and positive comments about our work. And we appreciate the reviewer for calling our attention on recent papers highlighting sex differences in cancers. We added an additional paragraph in the Discussion section describing sex differences in cancers with these recent papers covered. This additional discussion is now featured in the revised Main Text, lines 328-344.